# SPEECH WORLD MODEL: CAUSAL STATE–ACTION PLANNING WITH EXPLICIT REASONING FOR SPEECH

**Xuanru Zhou**[1,*] **Jiachen Lian**[2*]**, Henry Hong**[2]**, Xinyi Yang**[1] **& Gopala Anumanchipalli**[2]
[1]Zhejiang University, [2]UC Berkeley
`xuanruzhou15@gmail.com, jianchenlian@berkeley.edu`

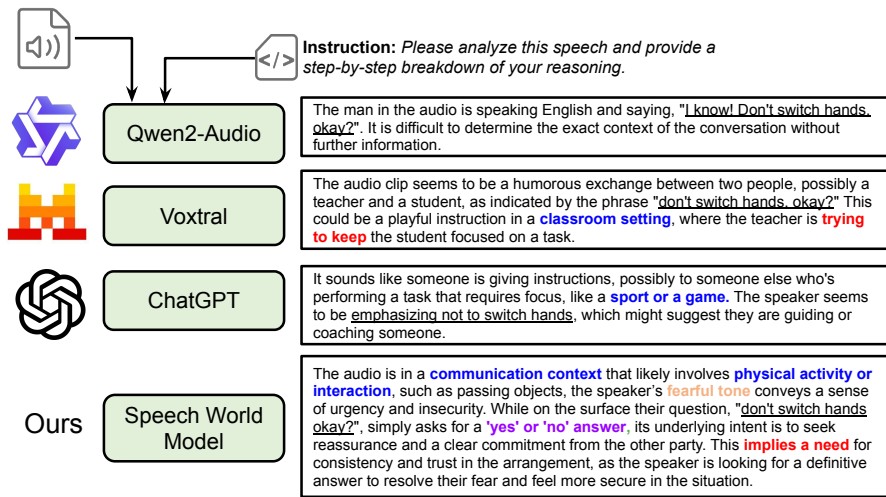

Figure 1: Speech World Model. Demo audio link: `http://bit.ly/4pBJuWP`.

## ABSTRACT

Current speech-language models (SLMs) typically use a cascade of speech encoder and large language model, treating speech understanding as a single black box. They analyze the content of speech well but reason weakly about other aspects, especially under sparse supervision. Thus, we argue for explicit reasoning over speech states and actions with modular and transparent decisions. Inspired by cognitive science we adopt a modular perspective and a world model view in which the system learns forward dynamics over latent states. We factorize speech understanding into four modules that communicate through a causal graph, establishing a cognitive state search space. Guided by posterior traces from this space, an instruction-tuned language model produces a concise causal analysis and a user-facing response, enabling counterfactual interventions and interpretability under partial supervision. We present the first graph based modular speech model for explicit reasoning and we will open source the model and data to promote the development of advanced speech understanding.

## 1 INTRODUCTION

Speech language models (SLMs) (Cui et al., 2024) provide a unified framework for general-purpose speech understanding, enabling multitask perception, reasoning, and multi-turn interaction. At the same time, an ongoing debate persists regarding whether such models truly exhibit genuine reasoning capabilities or merely perform sophisticated pattern matching and statistical abstraction over tokens (Shojaee et al., 2025). One direct observation is that current SLMs appear to combine outputs from isolated tasks—such as automatic speech recognition, emotion recognition, communicative

---

*Equal contribution.

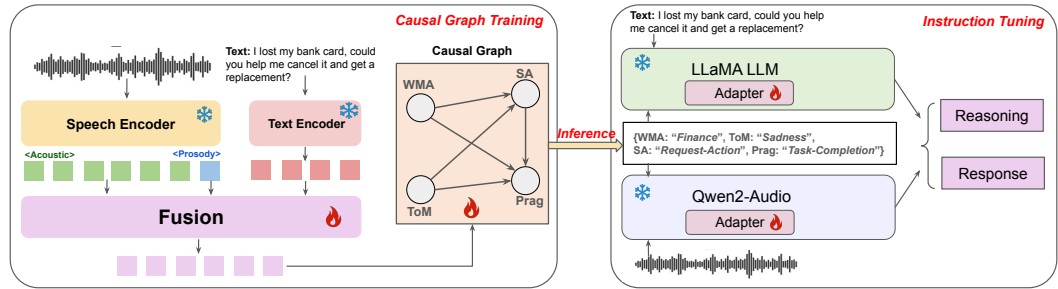

Figure 2: The *Speech World Model* pipeline with a running example, illustrating the "Causal Graph-Guided Explicit Reasoning" process. **(1) Causal Graph Training:** multimodal inputs (text $x$, acoustic $a$, prosody $z$) are encoded and fused to $g = \phi(h_x, h_a, h_z)$. Each node state $S_v$ is inferred from its parents $\text{Pa}(v)$ and fused feature $g$ via $S_v = \text{softmax}(f_v(g, \{S_u\}_{u \in \text{Pa}(v)}))$, yielding structured reasoning. **(2) Instruction Tuning:** these states are used as explicit guidance for the large (speech) language models to generate response $y$ by maximizing $\mathcal{L}_{\text{IT}} = -\sum \log P_\theta(y \mid \text{Instr}, x, S)$.

function detection, intent classification, and pragmatic inference—and then present this aggregation *as if it reflects reasoning*. However, this overlooks the *intrinsic dependencies among these speech components* and provides only partial supervision, which we argue is fundamental to enabling genuine reasoning in speech understanding. From a human-centered cognitive perspective, speech perception is thought to be hierarchically organized into partially specialized modules—such as acoustic, linguistic, and paralinguistic processing—that interact through causal and reciprocal relations (Hickok & Poeppel, 2007). However, this does not entail designing strictly brain-like modular systems (e.g. verified AI (Seshia et al., 2022)), as human perception itself operates through overlapping and partially opaque processes that remain incompletely understood (Angelaki et al., 2025).

An alternative line of research seeks an intermediate approach that models latent internal states, thereby enabling more interpretable predictions of system dynamics. A representative example is *World Models* (Ha & Schmidhuber, 2018; LeCun, 2022), which posit that the subsequent state is conditionally predictable from any given current state. In the domain of speech, human perceptual systems also anticipate interdependent dynamics across states such as emotion, intent, and linguistic content, such that changes in one dimension systematically constrain others (Hickok & Poeppel, 2007). These observations point to an underlying structure of causal dependencies. It is worth noting that Chain-of-Thought (CoT) prompting (Wei et al., 2022c) represents a parallel approach, which expands the search space to improve goal-directed reasoning at the expense of increased computational cost. However, the search process is not grounded in human speech perception principles.

In this work, we propose *Speech World Models (SWM)* to advance speech understanding and reasoning by modeling the dynamics of internal speech states. SWM comprises four explicit modules—*World Model Action (WMA)* (Mesaros et al., 2018), *Theory of Mind (ToM)* (Sclar et al., 2025), *Speech Act (SA)* (Bothe et al., 2020), and *Pragmatic Intent (Prag)* (Stolcke et al., 2000)—as an approximated subspace of the cognitive perception system. A causal-graph–based perception module is introduced to bridge these components and capture the flow of causal information. In the first stage, this graph is trained in both supervised and semi-supervised settings to establish a cognitive state search space. Notably, leveraging causal dependencies among modules yields faster convergence than training the modules independently. Moreover, because the graph approximates cognitive state structure, it can reliably infer unlabeled modules from partially labeled data, suggesting that causal graphs function as natural generative structures for latent variables. Once the search space is established, in the second stage we incorporate the outputs of the causal graph—samples from the search space, state nodes, and information flow—into prompts for instruction tuning. This provides an explicit reasoning advantage: by enforcing structured reasoning chains, SLMs are guided to reduce hallucinations and achieve greater overall reasoning capacity.

Experiments demonstrate the superiority of SWM over traditional SLMs across a range of reasoning metrics. First, when comparing against a randomly initialized graph without cognitive grounding, we find that our causal graph converges faster during training and achieves substantially higher reasoning scores (e.g., ACE, ICS). Second, instruction-tuning results show that SWM outperforms open-source SLMs such as Qwen1 (Chu et al., 2023), Qwen2-Audio (Chu et al., 2024), and Voxtral (Liu et al., 2025) in reasoning ability, with particularly strong performance in emotion recognition—even

surpassing commercial models—highlighting the benefits of explicit reasoning. Third, while the overall speech understanding performance of SWM (as measured by our proposed metrics) remains slightly below that of Gemini 2.5 Pro (Gemini Team, 2025), our training cost is dramatically lower (only 20 GPU hours), validating the efficiency of the causal-graph–based explicit reasoning paradigm.

## 2 RELATED STUDY

### 2.1 COGNITIVE FOUNDATIONS OF SPEECH PROCESSING

Cognitive neuroscience shows that speech perception and comprehension emerge from modular computations over latent states, rather than a single monolithic process. Core frameworks include the dual–stream model linking ventral (semantic) and dorsal (sensorimotor) pathways (Hickok & Poeppel, 2007), staged accounts of syntactic and semantic processing (Friederici, 2011), and analyses of the superior temporal gyrus that reveal intermediate phonological and prosodic representations (Leonard & Chang, 2022). Complementary perspectives highlight cortical oscillations aligned with hierarchical linguistic units (Giraud & Poeppel, 2012) and predictive coding views of the brain as a generative model updating latent states via prediction errors (Friston et al., 2021). Together, these perspectives converge on four principles: (i) modular and hierarchical decomposition of speech, (ii) forward predictive dynamics, (iii) integration of perception and action, and (iv) generative modeling of latent auditory states. Our Speech World Model operationalizes these principles computationally.

### 2.2 SPEECH LANGUAGE MODELS AND REASONING

Early work on speech language models (Gong et al., 2024; 2023; Tang et al., 2024; Kong et al., 2024) pioneered the use of instruction tuning pipelines for speech understanding and preliminary reasoning tasks. Subsequent studies adopted Chain-of-Thought (CoT) prompting (Wei et al., 2022c) to construct reasoning-chain search spaces, which advanced reasoning capabilities in speech-related tasks (Ma et al., 2025; Xie et al., 2025a;b; Kong et al., 2025); see also (Cui et al., 2024) for a broader review. Nevertheless, these CoT-based approaches remain largely heuristic and are not grounded in principles of human speech cognition (Hickok & Poeppel, 2007).

### 2.3 WORLD MODELS

The concept of world models has long-standing roots in cognitive science and neuroscience, where internal models are thought to support prediction, planning, and reasoning (Craik, 1967; Friston, 2005; Hickok & Poeppel, 2007). While recent computational advancements have successfully operationalized this concept via latent dynamics for planning and reinforcement learning (Ha & Schmidhuber, 2018; Hafner et al., 2019; Schrittwieser et al., 2020; LeCun, 2022), a parallel line of research emphasizes the *structured* nature of these models. Ullman & Tenenbaum (2020) and Lake et al. (2017) argue that robust learning resembles theory building, where world models internalize intuitive theories of physics and psychology. Instead of merely predicting sensory streams, such cognitive world models explicitly reason about causal mechanisms and agents' mental states (Baker et al., 2017). In the context of generative AI, this perspective inspires systems that not only model data distributions but also capture the underlying hierarchical structure of the domain, such as vision (Garrido et al., 2024) and language (Lin et al., 2024).

## 3 SPEECH WORLD MODEL SYSTEMS

In this section, we present the methodology behind our proposed *Speech World Model (SWM)*. At its core, the framework relies on a causal graph that factorizes speech understanding into modular components, enabling explicit reasoning over speech states and actions. These states then condition (spoken) language models, which are fine-tuned to generate both a transparent reasoning trace and a potential response to given utterance. This design allows us to validate the hypothesis that explicit state representation improves Speech Language Model's reasoning capacity and reduces hallucinations (Atwany et al., 2025).

### 3.1 WORLD MODEL CAUSAL GRAPH

We proposed a probabilistic causal model, represented by a Directed Acyclic Graph $\mathcal{G} = (\mathcal{V}, \mathcal{E})$, where $\mathcal{V}$ is a set of nodes representing different modules and $\mathcal{E}$ is a set of directed edges representing

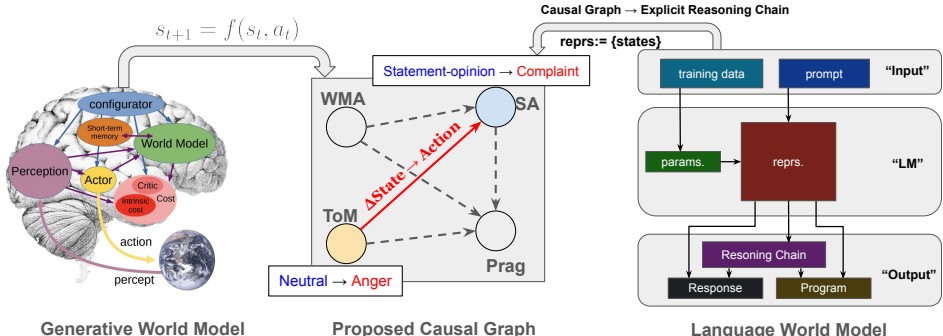

Figure 3: A unified perspective on world models. Both the Generative (left (Garrido et al., 2024)) and Language (right) world models are forward dynamic models. Our Causal Graph (center) provides a structured, explicit formulation of these dynamics. See Appendix A.1 for details.

causal relationships between modules. This structure allows us to explicitly model the dependencies between different aspects of speech and perform causal reasoning.

### 3.1.1 MODULES AND STATES

To construct a holistic casual graph for speech understanding, we select a set of latent variables that are both *necessary* and *sufficient* to describe the communicative process. Drawing from the standard model of communication (Shannon, 1948) and cognitive pragmatics (Bara, 2011), we define four key modules. These modules are not arbitrary selections but represent a hierarchical cognitive chain, progressing from situational grounding (Context) to agent modeling (Mental State), and finally to communicative action (Act) and goal (Intent). This combination ensures a comprehensive coverage of essential speech properties, spanning the "where, who, what, and why" to form a complete *cognitive perceptual subspace*. The complete label spaces are provided in Appendix A.4.2.

- **World Model Activation (WMA)** represents the *situational grounding* of the interaction. Speech does not occur in a vacuum but is situated in a specific functional context. Drawing on the concept of *Situation Models* (Zwaan & Radvansky, 1998), WMA captures the active domain or scenario (e.g., *SmartHome, Finance, Alarm*) (Bastianelli et al., 2020). This serves as the global precondition, "activating" the relevant knowledge partition required to interpret subsequent events.

- **Theory of Mind (ToM)** models the *agent's internal state*. Originating from cognitive science, ToM refers to the ability to attribute mental states (e.g. intents, emotions) (Premack & Woodruff, 1978; Baron-Cohen et al., 1985). While recent research has extensively explored ToM capabilities in Large Language Models, ranging from emergent behaviors (Kosinski, 2023) to robust reasoning evaluation (Sap et al., 2019; Sclar et al., 2025), we explicitly operationalize this concept for speech understanding. In our framework, this module captures the speaker's affective state and personality traits, acting as the internal driver for the observed speech behavior.

- **Speech Act (SA)** identifies the *communicative function* of the utterance. Based on the foundational Speech Act Theory by Austin (Austin, 1962) and Searle (Searle, 1969), this module categorizes the utterance into illocutionary acts (e.g., Questioning, Commanding, Asserting). It represents the surface-level "action" performed by the speaker (Bothe et al., 2020).

- **Pragmatic Intent (Prag)** represents the *underlying goal* or the perlocutionary effect the speaker aims to achieve. Pragmatics deals with the "unspoke" logic and implicature (Levinson, 1983). It captures *why* the speaker performed a specific act (e.g., using a question to sarcasm or to solicit help), grounded in the theory of Indirect Speech Acts (Ruytenbeek, 2021).

**Graph Perspective.** Each module corresponds to a node, instantiated as a neural network performing its own computation. Formally, we model the state of each module as a discrete categorical variable. The outputs of these modules are categorical latent states, and collectively they form the structured representation of speech, as presented in Fig. 3.

**World Model Perspective.** Inspired by the essence of image world models (Garrido et al., 2024), we view the entire system as a forward dynamics model: the current latent state, together with an

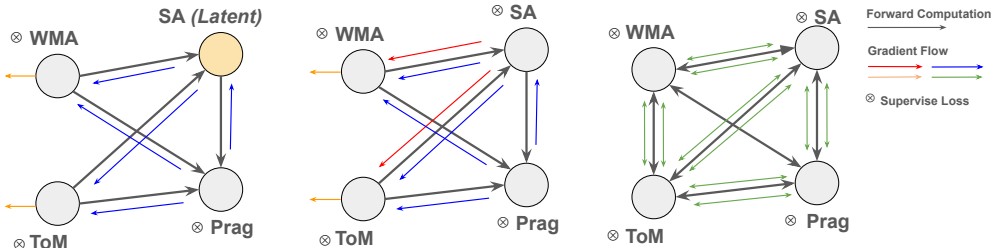

Figure 4: Comparison of gradient flow under different training scenarios.(A) Semi-supervised causal graph: when a module lacks direct supervision, it act as latent variable generator, and gradients propagate backward through causal edges from downstream modules. (B) Fully-supervised causal graph: all modules have labels, so losses are applied locally, while causal dependencies still guide gradient flow. (C) Fully-supervised random graph: supervision is present at every node but edges are unstructured, leading to redundant and less efficient gradient propagation.

action, leads to the next latent state. Within this view, the notion of action extends beyond external commands. Instead, an action can be understood as the causal influence exerted by one state on another. For example, when the Theory of Mind module shifts from *neutral* to *anger*, the downstream Speech Act may change from *Statement-opinion* to *Complaint/Escalate*, as shown in Fig. 3.

Thus, actions in our framework are best understood as the state transitions induced by causal relationships within the graph. Formally, for a node $v$ with parents $\text{Pa}(v)$:

$$S_v = f_v\big(\{S_u : u \in \text{Pa}(v)\}, A_{u \to v}\big), \tag{1}$$

where $S_v$ is the latent state at node $v$, and $A_{u \to v}$ denotes the action, interpreted as the causal influence (action) of parent node $u$ on $v$.

### 3.1.2 GRAPH CALCULATION

Given an input speech signal $X$, the casual graph infers a joint posterior distribution over latent states $Z = \{z_{WMA}, z_{ToM}, z_{SA}, z_{Prag}\}$. By the DAG factorization:

$$
\begin{aligned}
p(Z|X) = {} & p(z_{WMA}|X) \cdot p(z_{ToM}|X) \cdot \\
& p(z_{SA}|z_{WMA}, z_{ToM}, X) \cdot p(z_{Prag}|z_{SA}, z_{ToM}, z_{WMA}, X)
\end{aligned} \tag{2}
$$

Each conditional $p(z_v|\text{Pa}(v), X)$ is parameterized by a neural module $f_v(\cdot; \theta_v)$, modeled as a multi-class classifier. Our proposed casual graph is indicated in Fig. 4 (B). The detailed calculation and model architecture are detailed in Appendix A.2.1.

### 3.1.3 MULTI-TASK TRAINING WITH CASUAL GRAPH

Causal graph is trained in a multi-task manner to form reasoning-chain search space. The supervised loss is shown in Eq.3, where $m_{i,v}$ indicates whether the label for node $v$ is available in sample $i$.

$$\mathcal{L}_{\text{sup}} = \sum_{i=1}^{N} \sum_{v \in \mathcal{V}} m_{i,v} \, \text{CE}(y_{i,v}, S_{i,v}), \tag{3}$$

Teacher forcing (Bengio et al., 2015) is applied edge-wise: each child node receives either the ground-truth parent state or the predicted distribution. For edge $u \to v$:

$$\tilde{S}_{i,u} = \tau_{i,u \to v} \, \text{onehot}(y_{i,u}) + \big(1 - \tau_{i,u \to v}\big) \, \text{stopgrad}(S_{i,u}), \quad \tau_{i,u \to v} \sim \text{Bernoulli}(p_{u \to v}). \tag{4}$$

The mixed signal $\tilde{S}_{i,u}$ is used as the parent input when computing the state of child $v$.

### 3.1.4 SEMI-SUPERVISED LEARNING

**Motivation.** Annotations across modules are costly and often incomplete (e.g., WMA labels are missing more frequently). If parameter updates were restricted only to nodes with available labels, a large fraction of training data would be wasted. Semi-supervised training allows unlabeled parent nodes to be optimized through the supervision signals of their labeled children.

**Gradient Flow.** Consider a parent node $j$ without labels in sample $i$ ($m_{i,j} = 0$), where one of its child node $k$ has a label ($m_{i,k} = 1$). To ensure a differentiable path to the unlabeled parent node $j$, teacher forcing is disabled ($p_{j \to k}$=0), forcing the child $k$ to rely on the parent's continuous prediction $S_{i,j}$. Thus, the loss $\mathcal{L}_{i,k} = \mathrm{CE}(y_{i,k}, S_{i,k})$ propagates to $\theta_j$ through the chain rule:

$$\frac{\partial \mathcal{L}_i}{\partial \theta_j} = \sum_{\substack{k:\, m_{i,k}=1 \\ j \in \mathrm{Pa}(k)}} \frac{\partial \mathcal{L}_{i,k}}{\partial \eta_{i,k}} \frac{\partial \eta_{i,k}}{\partial S_{i,j}} \frac{\partial S_{i,j}}{\partial \eta_{i,j}} \frac{\partial \eta_{i,j}}{\partial \theta_j} \neq 0. \tag{5}$$

where the gradient also flows into the parent distributions $p(s_u \mid \xi_u)$. Thus unlabeled parents act as *latent variable generators* that receive updates from their supervised children as shown in Fig. 4 (A).

Multi-task supervision ensures "learn whenever a label is available" (Eq. 3), while semi-supervised training ensures "learn even when labels are missing" via chain gradients (Eq. 5). Together they allow effective learning from incompletely annotated data under the causal graph structure.

## 3.2 RANDOM GRAPH

To validate the benefits of our proposed causal structure, grounded by cognitive speech perception, we introduce the *Random Graph* as a strong, unstructured and data-driven baseline. This model replaces the predefined DAG with a fully connected graph where every module can, in principle, influence every other module. Consequently, the prediction for each module is conditioned on both the speech input $X$ and the initial, teacher-forced estimates from **all** other modules in the graph, allowing the model to learn arbitrary dependencies directly from the data.

$$p(Z|X) = \prod_{v \in \mathcal{V}} p(z_v | Z \setminus \{z_v\}, X) \tag{6}$$

By comparing this model with our structured causal graph, we can quantify the contribution of imposing theoretically-grounded causal priors on speech understanding. The detailed, step-by-step computation for this iterative process is provided in Appendix A.2.2.

## 3.3 SPEECH UNDERSTANDING AND REASONING

To bridge the gap between the structured latent states inferred by the causal graph and human-interpretable analysis, we employ an instruction tuning (Wei et al., 2022a) phase. This final stage is designed to validate our core hypothesis: *Conditioning on these explicit graph states tends to guide the reasoning-chain search toward human-aligned spaces.*. To this end, the model is trained to produce a transparent reasoning process and a contextually appropriate response based on the graph's outputs. We explore two distinct settings for this purpose: a language-only approach and a multi-modal approach.

### 3.3.1 LANGUAGE-ONLY SETTING

In this setting, we fine-tune a standard LLM to reason over the symbolic outputs of the pre-trained causal graph $\mathcal{G}$. The input is prompted with a general instruction (Instr) and a textual serialization of the graph's output, $I(\mathcal{G}(x))$, which encodes the inferred states and their causal relationships. The prompts we used for training are detailed in Appendix. A.5.2.

The model is trained to produce a target sequence y that contains both a step-by-step *reasoning process* and a potential *dialogue response*. The training objective is to minimize the standard cross-entropy loss over the target sequence, where $\theta$ represents the parameters of the LLM, and the target $y$ is the ground-truth reasoning and response pair.

$$\mathcal{L}_{\mathrm{IT}}(\theta) = - \sum_{(x,y) \in \mathcal{D}} \log p_\theta \left( y \,\middle|\, \mathrm{Instr},\, I(\mathcal{G}(x)) \right) \tag{7}$$

### 3.3.2 MULTI-MODAL SETTING

This approach leverages a speech language model to directly ground the symbolic states in the acoustic properties of the raw speech signal. The input to the SLM consists of the instruction (Instr),

the raw speech input $x$, and the set of inferred states $\{S_v\}_{v \in \mathcal{V}}$. The model's task is to generate the same target sequence $y$ containing a reasoning process and a response.

$$\mathcal{L}_{\text{IT}}(\theta) = - \sum_{(x,y) \in \mathcal{D}} \log p_\theta \Big( y \,\Big|\, \text{Instr}, x, S_{\text{WMA}}, S_{\text{ToM}}, S_{\text{SA}}, S_{\text{Prag}} \Big) \tag{8}$$

This objective forces the SLM to build a joint understanding of the audio signal and the structured states, effectively learning to associate "what was said" and "how it was said" with the "why" provided by our causal graph's analysis.

## 4 EXPERIMENTS

### 4.1 REAL-WORLD DATASET

The following four datasets serve as the primary speech sources for our experiments. Please refer to Appendix A.4.1 for detailed statistics regarding these datasets.

- **MELD** (Poria et al., 2019) is a conversational dataset for emotion recognition. It consists of over 13,000 utterances from the TV series Friends, annotated with emotion and sentiment labels.
- **IEMOCAP** (Busso et al., 2008) is an audiovisual and multi-speaker database of dyadic interactions between actors, containing approximately 12 hours of data annotated with categorical and dimensional emotion labels.
- **SLURP** (Bastianelli et al., 2020) is a challenging spoken language understanding (SLU) dataset featuring single-turn user interactions with a voice assistant. Utterances are annotated with a hierarchical schema, providing both user intent and corresponding action.
- **VoxCeleb** (Nagrani et al., 2017) is a large-scale text-independent speaker identification dataset. It comprises hundreds of thousands of utterances from celebrities, extracted from YouTube videos.

### 4.2 LABEL-GENERATION

A central challenge for scaling up is that real-word speech data usually comes with only *partial supervision*: some modules may be annotated while others are not. To construct complete instances for fully-supervised graph training and instruction tuning, we implemented a robust two-stage label generation pipeline leveraging **Vicuna-13b-v1.5** (Chiang et al., 2023) as the teacher model.

**Stage 1: Constrained Label Imputation.** Given the speech transcription and any available module labels, the goal is to infer missing labels. We employ a one-shot prompting strategy (Brown et al., 2020), providing the model with the complete label space definition and a single demonstrative example to enforce the desired output format. To ensure the generated labels map strictly to valid categories, we apply a constrained decoding mechanism that maps the generated tokens to the nearest canonical label by minimizing the *Levenshtein edit distance*. As a quality control measure, instances where the model fails to generate parsable or valid labels after matching are discarded.

**Stage 2: Reasoning and Response Synthesis.** By leveraging the robust causal priors inherent in LLMs (Kiciman et al., 2024; Binz & Schulz, 2023), we explicitly aim to simulate the latent data-generating process of speech. Conditioned on the complete labels (from stage 1) and transcription, we query the LLM again to synthesize the downstream instruction data: (i) a comprehensive reasoning analysis which explains how the inferred states interact through the causal graph to shape the utterance, and (ii) a contextually appropriate response aligned with these states.

Full details of prompts and other implementation specifics are provided in the Appendix A.4.2.

### 4.3 METRICS

**Node quality.** We evaluate the performance of individual modules (nodes) on their respective sub-tasks using standard classification *Accuracy* and *weighted F1-Score*.

**Edge causal effect.** To validate each learned edge $X \to Y$, we measure its causal strength using the *Average Causal Effect (ACE)* (Pearl, 2010) and its directional consistency with the *Intervention Consistency Score (ICS)*. *ACE* indicates the magnitude of post-intervention effects, while *ICS* indicates if effects align with a predefined hypothesis. See Appendix A.8.1 for computational details.

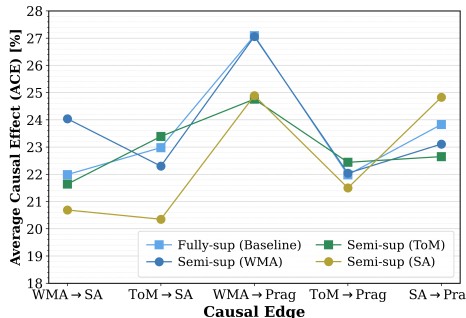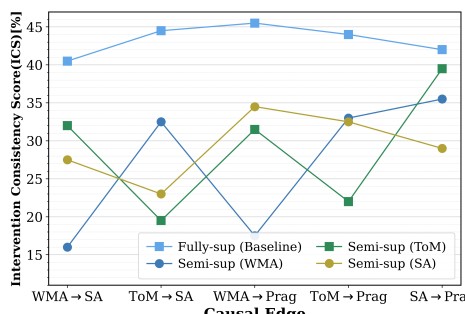

Figure 5: *ACE* and *ICS* of each casual edge under both fully-supervised and semi-supervised training.

**Instruction tuning.** To provide a scalable and consistent evaluation of our instruction-tuned models, we employ the Model-as-Judge (M.J.) methodology (Zheng et al., 2023), where GPT-4o (OpenAI, 2024) acts as an impartial judge to score both reasoning and response quality. The overall M.J. score is a weighted sum of reasoning and response scores. *EM* and *EA* refer to emotion mention rate and emotion classification accuracy, respectively, while *R-Len* represents the length of the reasoning process. The detailed prompting strategy and scoring rubric are indicated in Appendix A.9.

## 4.4 EXPERIMENTS SETUP

Our method employs a two-stage training pipeline: causal graph training followed by instruction tuning. The causal graph consists of MLP-based classifiers with temporal attention, trained under fully- and semi-supervised settings. To validate training efficiency, we report wall-clock times averaged over 3 seeds (42, 123, 2025) on a single NVIDIA A6000 GPU. Our Causal Graph converges in 2.07h ($\pm$ 0.04), achieving a $\sim$5$\times$ speedup compared to the Random Graph baseline (10.39h $\pm$ 0.15) under identical settings. For instruction tuning, we apply LoRA (Hu et al., 2022) to both Llama3.1-8B Meta (2024) (language-only) and Qwen2-Audio (Chu et al., 2024) (multi-modal). Training on 4 NVIDIA A6000 GPUs required 19h for the language-only model and 24.6h for the multi-modal model, respectively. We benchmark against a Qwen2-Audio baseline fine-tuned with CoT prompts. Full architectures, configurations, and ablations are detailed in Appendices A.5, A.7, and A.6.

## 4.5 GRAPH EVALUATION

The results of our graph evaluation are presented in Table. 1 and Table. 2, with a detailed breakdown of edge causal effects shown in Fig. 5. These results first demonstrate the **efficiency** of our Causal Graph, which **achieves node accuracy** comparable to a Random Graph baseline at a significantly **lower training cost**. Table 2 exposes a fundamental flaw in the Random Graph: **structural instability**. The dominant information flow in the Random Graph fluctuates chaotically with training hyperparameters (teacher-forcing ratio), suggesting it functions as a black-box that exploits *spurious correlations* rather than learning invariant mechanisms. In contrast, our Causal Graph maintains consistent ACE and ICS scores, proving it captures stable, interpretable causal dependencies. Fig. 5 offers a granular view of model's robustness under partial supervision. Crucially, when a specific module (e.g., ToM) is unsupervised, the drop in ACE is logically **localized** to the edges connected to that node (e.g., ToM→SA), while other pathways (e.g., WMA→SA) remain unaffected. This **orthogonality** confirms that our modules have learned disentangled representations, allowing the system to effectively infer missing states via the causal structure rather than collapsing.

## 4.6 SPEECH UNDERSTANDING AND REASONING EVALUATION

Our speech understanding and reasoning results (Table. 3) reveal the distinct contributions of our reasoning framework. To isolate these effects, we created a tuned baseline by fine-tuning Qwen2-Audio with our CoT-style instruction data. This **baseline alone surpasses other open-source models**, confirming

Table 2: Node quality and information flow analysis for random graph baseline. (Detailed in Appendix A.8.2)

| $p$(teacher_force) | Node Quality (Accuracy %, ↑) | | | | Information-flow | |
| | WMA | ToM | SA | Prag | Strongest | Weakest |
|---|---|---|---|---|---|---|
| 0.0 | 70.1 | 75.2 | 66.1 | 82.8 | SA → WMA | WMA → Prag |
| 0.3 | 69.7 | 74.0 | 67.5 | 83.6 | ToM → SA | ToM → Prag |
| 0.5 | 69.5 | 73.1 | 65.9 | 82.0 | WMA → SA | ToM → Prag |
| 0.8 | 69.4 | 74.4 | 70.3 | 83.2 | ToM → Prag | Prag → ToM |
| 1.0 | 70.1 | 73.9 | 66.3 | 82.3 | SA → ToM | ToM → Prag |

the high quality of our instruction tuning data, and our full SWM models, which leverage the causal

Table 1: Performance evaluation of the causal graph on node accuracy and edge validity under different supervision settings. The gray background in the semi-supervised rows highlights the accuracy of modules that were left unlabeled during training, demonstrating the model's ability to infer latent states via causal structure. The rightmost columns evaluate the strength of learned causal dependencies using verage Causal Effect (ACE) and Intervention Consistency Score (ICS).

| Method | Setting | Node Quality (Accuracy %, ↑) | | | | Edge Casual Effect | |
|--------|---------|------|-----|-----|------|----------------|----------------|
| | | WMA | ToM | SA | Prag | Ave. ACE (%, ↑) | Ave. ICS (%, ↑) |
| Fully-supervised | – | 69.4 | 73.5 | 65.3 | 81.4 | 23.57 | 43.29 |
| Semi-supervised | Latent module: WMA | 34.8 | 75.0 | 70.7 | 83.2 | 21.71 | 26.9 |
| | Latent module: ToM | 69.1 | 43.3 | 69.6 | 83.5 | 21.98 | 28.9 |
| | Latent module: SA | 69.3 | 77.0 | 34.4 | 82.5 | 21.65 | 29.3 |
| Random Graph | – | 69.7 | 74.0 | 67.5 | 83.6 | – | – |

graph for explicit reasoning, **significantly outperform this tuned baseline**. This demonstrates that while good data is important, the causal graph's explicit guidance is the critical factor driving superior reasoning ability. This benefit is particularly stark in **emotion recognition**, where our model's accuracy exceeds even top proprietary models. We posit that the graph serves as a **structural anchor**, facilitating the **explicit disentanglement** of affective states (ToM), which effectively mitigates the common "text-dominance" bias and thereby reducing hallucination. Finally, while a model like Gemini 2.5 Pro (Gemini Team, 2025) holds an edge in the overall score, **our approach achieves this competitive performance at a dramatically lower training cost**. This demonstrates that integrating **structured cognitive priors** allows smaller models to punch above their weight, offering a resource-efficient alternative to the prevailing reliance on massive parameter scaling.

Table 3: Performance comparison against open-source and proprietary baselines. We report results under both Direct and CoT prompting styles. The Overall M.J. Score is the primary metric, calculated as a weighted aggregate ($0.6 \times R_s + 0.4 \times R_p$) to balance the Reasoning Score ($R_s$) and the final Response Score ($R_p$). The Reasoning Breakdown columns provide granular metrics, where *EM* and *EA* denote emotion mention rate and emotion classification accuracy, respectively. R-Len indicates the average length of the generated response in words.

| Method | Prompt Style | Overall M.J. Score ↑ | Reasoning Score ↑ | Response Score ↑ | Reasoning Breakdown (%) ↑ | | R-Len |
|--------|-------------|---------------------|-------------------|------------------|-------|-------|-------|
| | (Inference) | $0.6 \times Rs + 0.4 \times Rp$ | $Rs$ | $Rp$ | *EM* | *EA* | *(words)* |
| *Our Models* | | | | | | | |
| SWM (Llama3.1-8b) | CoT | **7.81** | **7.84** | 7.76 | **97.80** | 66.26 | **105.70** |
| SWM (Qwen2-Audio) | CoT | 7.59 | 7.26 | **8.08** | 91.80 | **71.02** | 104.64 |
| *Tuned Baseline* | | | | | | | |
| Qwen2-Audio-CoT | CoT | 5.18 | 4.76 | 5.82 | 92.11 | 34.72 | 102.44 |
| *Baselines* | | | | | | | |
| Qwen-Audio (Chu et al., 2023) | Direct | 2.70 | 2.20 | 3.46 | 14.20 | 8.00 | 28.99 |
| Qwen2-Audio (Chu et al., 2024) | Direct | 2.63 | 2.08 | 3.47 | 5.14 | 15.38 | 15.60 |
| Qwen2-Audio (Chu et al., 2024) | CoT | 2.39 | 1.96 | 3.04 | 6.11 | 17.50 | 21.19 |
| Voxtral (Liu et al., 2025) | Direct | 2.89 | 2.46 | 3.54 | 10.28 | 5.88 | 69.64 |
| Voxtral (Liu et al., 2025) | CoT | 2.92 | 2.52 | 3.52 | 10.89 | 5.56 | 71.42 |
| *Proprietary Models* | | | | | | | |
| GPT-4o (OpenAI, 2024) | CoT | 7.41 | 6.98 | 8.06 | 68.20 | 45.16 | 105.23 |
| Gemini 2.5 Pro (Gemini Team, 2025) | CoT | **8.12** | **8.02** | **8.28** | 82.47 | 51.29 | 112.62 |

## 5 LIMITATIONS AND CONCLUSIONS

In this work, we proposed Speech World Model (SWM), which surpasses the current open-sourced speech language models in understanding and reasoning ability by modeling the dynamics of internal speech states. However, there are still some limitations. First, the current model includes only four modules, while more diverse states could enhance speech understanding and better capture complex speech dynamics. Second, the causal graph in SWM is predefined, limiting the model's ability to adapt to unseen dependencies. Future work could focus on learning adaptive casual structures to increase flexibility. Third, while instruction tuning effectively integrates reasoning traces and responses, it relies heavily on the label generation pipeline, where inaccuracies can propagate to both the reasoning and response stages. Therefore, achieving efficient data annotation and generation for speech remains a crucial and collaborative challenge within the speech community.

ACKNOWLEDGEMENTS

We thank Anastasis Stathopoulos, Alexander H. Liu and Alexei Baevski for their valuable advice and support throughout this work.

ETHICS STATEMENT

Our work adheres to the ICLR Code of Ethics. All experiments were conducted on publicly available datasets, ensuring data privacy and consent. Our research is intended for positive applications, and we strongly discourage any misuse of this technology for surveillance or manipulative purposes.

REPRODUCIBILITY STATEMENT

To ensure the reproducibility of our results, we will make our source code, including implementation, training, and evaluation scripts, publicly available at `https://github.com/eureka235/eureka235.github.io`. All datasets used are public and cited accordingly. The main experimental setup is described in Sec. 4. For a more detailed breakdown of the model architecture, hyperparameters, and implementation specifics, please refer to Appendix. A.4 - A.9.

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

# A    APPENDIX

## A.1    DISCUSSION ON TWO TYPES OF WORLD MODEL

As stated in Fig. 3, we consider both generative world models, such as those conceptualized by LeCun (LeCun (2022)), and large language models as fundamentally being **forward dynamic models**. This perspective helps unify their roles and clarify the contribution of our proposed Causal Graph.

**Generative World Model.**    This model is often associated with agent-based learning, aims to learn a transition function of the environment. This function, typically formalized as $s_{t+1} = f(s_t, a_t)$, allows an agent to predict future sensory inputs (the next state $s_{t+1}$) based on the current state ($s_t$) and a chosen action ($a_t$). The learned representation $s_t$ is often a low-dimensional, latent state that captures the underlying dynamics of the world, enabling the agent to simulate outcomes and plan.

**Language World Model.**    This model can be viewed as a forward model operating in a symbolic space. Its core function is to predict the next token ($w_{t+1}$) given a history of previous tokens ($w_{1..t}$). This autoregressive process, $P(w_{t+1}|w_{1..t})$, is conceptually parallel to predicting the next state. Here, the "state" is the context provided by the preceding text, and the "action" is the generation of the next token. This process unfolds a trajectory through the space of language. Techniques like Chain-of-Thought (CoT) make this explicit, where the model generates a sequence of intermediate thoughts, effectively simulating a trajectory through a conceptual state space to reach a conclusion.

**Causal Graph.**    While the generative model learns implicit, sub-symbolic dynamics of an environment, the language model executes explicit, symbolic reasoning. Recognizing their shared foundation as forward models, our proposed Causal Graph offers a third, structured approach to represent these dynamics. It distills the world's complex dynamics into an explicit graph of causal relationships between key cognitive states (WMA, ToM, etc.). This structured knowledge then provides a powerful prior for the language model. Instead of navigating the vast space of possible text sequences based only on statistical patterns, the language model's forward-reasoning process is guided and constrained by the causal dynamics provided by our graph, leading to more grounded and coherent outputs.

## A.2    DETAILED GRAPH CALCULATIONS

### A.2.1    CAUSAL GRAPH

Let $x$ denote the transcription of given speech, $a$ the acoustic feature (WavLM (Chen et al., 2022)), and $z$ the prosodic feature (Eyben et al., 2010). We define text feature with distil-BERT encoder (Sanh et al., 2020):

$$h_{\text{text}} = E_{\text{text}}(x) \tag{9}$$

An optional pre-fusion operator $\phi$ (attention/gated/transformer) combines these with prosodic features:

$$g = \phi(h_{\text{text}}, z, a). \tag{10}$$

We consider four categorical latent states:

$$S_{\text{WMA}} \in \Delta^{C_{\text{wma}}-1}, \quad S_{\text{ToM}} \in \Delta^{C_{\text{tom}}-1}, \quad S_{\text{SA}} \in \Delta^{C_{\text{sa}}-1}, \quad S_{\text{Prag}} \in \Delta^{C_{\text{prag}}-1}, \tag{11}$$

each represented by a probability simplex.

For any node $v$ with parents $\text{Pa}(v)$, we compute its state by a node-specific neural map $f_v$ from the parents' distributions and available features:

$$S_v = f_v\big(\{S_u : u \in \text{Pa}(v)\}, \xi_v\big), \tag{12}$$

where $\xi_v$ denotes the feature bundle used by node $v$:

$$\xi_{\text{WMA}} = (h_{\text{text}}, a), \quad \xi_{\text{ToM}} = g, \quad \xi_{\text{SA}} = (g \text{ or } h_{\text{text}}), \quad \xi_{\text{Prag}} = (g \text{ or } h_{\text{text}}).$$

Concretely, each node produces logits and then a softmax:

$$S_v = \text{softmax}\big(W_v \cdot \psi_v([\xi_v, \{S_u\}_{u \in \text{Pa}(v)}])\big), \tag{13}$$

where $\psi_v(\cdot)$ is a nonlinear transformation and $W_v$ is a task-specific projection.

### A.2.2 RANDOM GRAPH

This architecture serves as an ablation, testing whether a model with the flexibility to learn arbitrary dependencies from data can outperform our proposed causal structure.

Instead of a uni-directional information flow, the Random Graph employs an iterative refinement process. The computation unfolds over two main iterations. Let $S_v^t$ be the probability distribution (state) of module $v \in \mathcal{V}$ at iteration $t$, and let $H$ represent the fused input features derived from the speech-text pairs, i.e., $H = g(X)$.

**Initialization (t=0).** At the initial step, the states of all peer modules are unknown and are represented by zero vectors.

$$S_u^{(0)} = \mathbf{0}, \quad \forall u \in \mathcal{V} \tag{14}$$

**First Iteration (t=1).** Each module $v$ computes an initial state estimate in parallel, conditioned on the input features $H$ and the zero-initialized states of all other modules. This step allows each module to form a preliminary prediction based solely on the primary input modalities.

$$S_v^{(1)} = f_v\left(H_{in}, S_u^{(0)} : u \in \mathcal{V}, u \neq v\right) \tag{15}$$

Here, $f_v$ is the neural network parameterizing module $v$. The output $S_u^{(1)}$ is the softmax-normalized probability distribution over the module's labels.

**State Refinement with Teacher Forcing.** Before the second iteration, we generate a refined input state $\tilde{S}_u^{(1)}$ for each module $u$. As in the causal graph training (Eq. 4), this is a stochastic mixture of the predicted states $S_u^{(1)}$ and the ground-truth label $y_{i,u}$ via teacher forcing. This allows the model to correct initial errors and propagate more accurate information in the next step.

**Final Iteration (t=2).** In the final step, each module computes its output by conditioning on the input $H$ as well as the refined states $\tilde{S}_u^{(1)}$ from all other modules. This allows for a form of message-passing where modules can adjust their predictions based on the initial estimates of their peers.

$$S_v^{(\text{final})} = f_v\left(H_{in}, \tilde{S}_u^{(1)} : u \in \mathcal{V}, u \neq v\right) \tag{16}$$

The final logits from $S_v^{(\text{final})}$ for all modules are then used to compute the multi-task loss as defined in Eq. 3. By comparing this model with our structured causal graph, we can quantify the contribution of imposing theoretically-grounded causal priors on the task of speech understanding.

### A.3 THEORETICAL ANALYSIS OF THE SPEECH WORLD MODEL

In this section, we provide a rigorous analysis of how the causal structure of the Speech World Model enhances training efficiency compared to unstructured baselines. We analyze this improvement through two theoretical lenses: (1) the optimization of gradient flow via structural priors, and (2) the pruning of the search space for downstream instruction tuning.

#### A.3.1 STRUCTURED PRIORS AND OPTIMIZED GRADIENT FLOW

The core advantage of Speech World Model lies in modeling the joint distribution over latent variables as a factorized probability, where each module is conditioned only on its causal parents (Pearl, 2009):

$$P(Y_1, \ldots, Y_N | X) = \prod_{i=1}^{N} P(Y_i | X, Y_{\mathrm{Pa}(i)}) \tag{17}$$

To rigorously quantify the efficiency gains afforded by this factorization, we analyze the gradient propagation dynamics, contrasting our structured approach with a standard monolithic architecture.

**Gradient Flow and Redundancy in Monolithic Models.** Consider a monolithic baseline where the latent states $\{S_1, \ldots, S_N\}$ are fully interconnected. The state of any module $S_i$ is potentially a function of all other states: $S_i = f_i(S_1, \ldots, S_{i-1}, S_{i+1}, \ldots, S_N, X)$. Let the total loss be $\mathcal{L} = \sum_{i=1}^{N} \lambda_i \mathcal{L}_i(S_i)$, where $\lambda_i \in \{0, 1\}$ indicates label availability. The gradient with respect to the parameters $\theta_k$ of a single module $k$ expands via the chain rule into a dense sum over all paths:

$$\frac{\partial \mathcal{L}}{\partial \theta_k} = \sum_{i=1}^{N} \lambda_i \left( \sum_{j=1}^{N} \frac{\partial f_i}{\partial S_j} \frac{\partial S_j}{\partial \theta_k} + \frac{\partial f_i}{\partial S_k} \frac{\partial S_k}{\partial \theta_k} \right) \tag{18}$$

In this fully connected setting, the term $\frac{\partial S_i}{\partial \theta_k}$ is non-trivial for all pairs. The model must implicitly learn to "prune" irrelevant connections by driving the corresponding Jacobians $\frac{\partial f_i}{\partial S_j}$ to zero, which is computationally inefficient and data-intensive. We formally define this wasted optimization effort as *learning redundancy*:

$$\text{Redundancy} \propto \sum_{i=1}^{N} \Pi_i \left\| \nabla_{\theta_i} \mathcal{L}(\theta_i) \right\|_{\text{non-causal}} \tag{19}$$

where $\Pi_i$ serves as an importance indicator. High redundancy implies the model expends gradient updates discovering structural independencies that could have been provided as priors.

**Sparse Gradients in the Causal Model.** Our Speech World Model eliminates this redundancy by construction. Since the state $S_i$ depends strictly on its parents $S_{\mathrm{Pa}(i)}$, the system's Jacobian matrix becomes sparse, satisfying $\frac{\partial S_i}{\partial S_j} = 0$ if $j \notin \mathrm{Pa}(i)$. Consequently, the gradient calculation for parameters $\theta_k$ simplifies dramatically: $\theta_k$ only influences the loss of module $j$ if $k$ is an ancestor of $j$ (denoted $k \in \mathrm{An}(j)$). The total gradient becomes:

$$\frac{\partial \mathcal{L}}{\partial \theta_k} = \lambda_k \frac{\partial \mathcal{L}_k}{\partial \theta_k} + \sum_{j \in \mathrm{Desc}(k)} \lambda_j \frac{\partial \mathcal{L}_j}{\partial \theta_k} \tag{20}$$

Unlike the monolithic case, each term $\frac{\partial \mathcal{L}_j}{\partial \theta_k}$ here is computed along a unique, valid causal path. By explicitly setting non-causal dependencies to zero, the model precludes the need to discover relationships from scratch. This factorization focuses updates strictly along causally relevant pathways, directly minimizing the redundancy term and accelerating convergence, which aligns with the established advantages of causality-inspired explainability (Schölkopf et al., 2021).

#### A.3.2 CONSTRAINED SEARCH SPACE FOR INSTRUCTION TUNING

The causal graph provides a critical advantage by constraining the combinatorially large search space of latent cognitive states, $\mathcal{Y} = \mathcal{Y}_1 \times \cdots \times \mathcal{Y}_N$, to a valid, low-entropy subspace $\mathcal{S}_{\mathcal{G}} \subset \mathcal{Y}$. This

reduction in complexity is quantifiable by information entropy:

$$H(Y|X, \mathcal{G}) = \sum_{i=1}^{N} H(Y_i|Y_{\mathrm{Pa}(i)}, X, \mathcal{G}) \ll \sum_{i=1}^{N} \log |\mathcal{Y}_i| \tag{21}$$

This simplification transforms the learning task for the language model. Existing reasoning frameworks, such as Chain-of-Thought (Wei et al., 2022b) and Tree-of-Thoughts (Yao et al., 2023), require the the model to navigate the vast state space $\mathcal{Y}$, either linearly or via tree-search, to approximate the joint distribution $\pi_\phi(Y, R|X)$. This implies a computationally expensive search for valid reasoning chains from scratch. Our approach simplifies this to learning a conditional policy $\pi_\phi(R|X, Y_\mathcal{G})$. By explicitly providing the causally pruned state $Y_\mathcal{G}$, we introduce a *control variate* that reduces gradient variance, thereby accelerating convergence and improving sample efficiency.

The improvement in sample efficiency discussed above can be rigorously derived by analyzing the variance of the gradient estimator used in fine-tuning.

**Instruction Tuning as Policy Gradient.**   Instruction tuning can be framed as learning a policy $\pi_\phi(A|C)$ that produces a response $A$ given a context $C$. The learning objective is to maximize the expected log-likelihood of a gold-standard response $A^*$:

$$J(\phi) = \mathbb{E}_{(C,A^*)\sim\mathcal{D}}[\log \pi_\phi(A^*|C)] \tag{22}$$

This objective is typically optimized via stochastic gradient ascent. The efficiency of this learning process is highly dependent on the variance of the gradient estimator, $\hat{g} = \nabla_\phi \log \pi_\phi(A^*|C)$. A lower variance estimator implies that fewer samples are needed to approximate the true gradient direction effectively.

**Variance in Standard Chain-of-Thought.**   In standard CoT, the context is the input query, $C = X$. The LLM must generate both the intermediate reasoning steps (the chain of thought, $Y$) and the final answer ($R$), so the action is $A = (Y, R)$. The gradient estimator is thus $\hat{g}_{\mathrm{CoT}} = \nabla_\phi \log \pi_\phi(Y^*, R^*|X)$. The variance of this estimator is high because for any given query $X$, there can be a multitude of valid reasoning paths $\{Y^*\}$ leading to the correct answer $R^*$. The model receives a noisy learning signal as it may be penalized for producing a perfectly valid reasoning chain that does not exactly match the single ground-truth example.

**Variance Reduction via Causal Conditioning.**   Our method fundamentally modifies the context $C$. By first executing the causal graph, we generate a structured, low-entropy state vector $Y_\mathcal{G} \sim P(Y|X, \mathcal{G})$. The language model is then conditioned on this richer context, $C' = (X, Y_\mathcal{G})$, simplifying its task to generating only the final response $R$. The policy becomes $\pi_\phi(R|X, Y_\mathcal{G})$ and the gradient estimator is $\hat{g}_{\mathrm{SWM}} = \nabla_\phi \log \pi_\phi(R^*|X, Y_\mathcal{G})$.

We can formally attribute the efficiency gain to the law of total variance:

$$\mathrm{Var}(\hat{g}_{\mathrm{CoT}}) = \underbrace{\mathbb{E}_{Y_\mathcal{G}}[\mathrm{Var}(\hat{g}_{\mathrm{SWM}}|Y_\mathcal{G})]}_{\text{Our Variance}} + \underbrace{\mathrm{Var}_{Y_\mathcal{G}}[\mathbb{E}(\hat{g}_{\mathrm{CoT}}|Y_\mathcal{G})]}_{\text{Variance Reduced}} \tag{23}$$

The gradient estimator in our model, $\hat{g}_{\mathrm{SWM}}$, corresponds to the inner conditional expectation term. Since we condition on the explicit information provided by $Y_\mathcal{G}$, our observed variance is limited to the first term. The second term, $\mathrm{Var}_{Y_\mathcal{G}}[\mathbb{E}(\hat{g}_{\mathrm{CoT}}|Y_\mathcal{G})]$, represents the variance arising from the ambiguity of latent cognitive states. Because $Y_\mathcal{G}$ is highly predictive of the final answer $R^*$, conditioning on it effectively "explains away" this large source of variance. As hypothesized, the provided state $Y_\mathcal{G}$ acts as a **control variate**, significantly reducing the gradient variance and directly leading to the greater sample efficiency observed in our experiments.

### A.4 DATA

#### A.4.1 DATASET STATISTIC

We construct our training dataset by aggregating four publicly available real-world datasets, as detailed in Table. 4. These datasets provide a diverse range of labels, which we leverage to train different modules of our Causal Graph. Specifically, we use the "Emotion" labels from MELD and IEMOCAP as ground truth for our Theory of Mind (ToM) module, and the "Intention," "Action," and "Scene" labels from SLURP for the Pragmatics (Prag) and World Model Activation (WMA) modules. For any remaining labels within a given dataset that are not directly used, we employ label generation to create pseudo-labels for the corresponding modules. For evaluation, we adhere to the official test set splits provided by the MELD and SLURP datasets. For IEMOCAP and the VoxCeleb subset, we randomly sample 10% of the data to create our test sets, ensuring no overlap with the training data.

Table 4: Statistics for Real-world Dataset.

| Dataset | # Samples | # Hours | Label | Source |
|---|---|---|---|---|
| MELD (Poria et al., 2019) | $\sim 13{,}000$ | $\sim 13$ | Emotion | TV series Friends |
| IEMOCAP (Busso et al., 2008) | $\sim 10{,}000$ | $\sim 12$ | Emotion | Dyadic actor dialogs |
| SLURP (Bastianelli et al., 2020) | $\sim 72{,}000$ | $\sim 58$ | Intention, Action, Scene | Voice assistant |
| VoxCeleb (subset) Nagrani et al. (2017) | $\sim 30{,}000$ | $\sim 30$ | Speaker identity | Youtube |
| Total | $\sim 125{,}000$ | $\sim 113$ | – | – |

#### A.4.2 LABEL GENERATION

The complete label space for each of the four modules in our Causal Graph: World Model Activation (WMA), Theory of Mind (ToM), Speech Act (SA), and Pragmatics (Prag) is detailed in the box below.

---

**Label Space for Causal Graph Modules**

**WMA:** Alarm, Calendar, Contacts, Communication, SocialMedia, Email, Music, Radio, Podcasts, Audiobooks, Games, Weather, News, Traffic, Transport, Shopping, FoodAndDrink, Cooking, SmartHome, IoTDevices, Cleaning, Finance, Stock, Conversion, Knowledge, Math, Definition, Chitchat, Recommendation, GeneralControl.

**ToM:** Neutral, Joy, Sadness, Anger, Surprise, Fear, Disgust.

**SA:** Statement-non-opinion, Statement-opinion, Yes-No-Question, Wh-Question, Acknowledge (Backchannel), Action-directive, Conventional-closing, Appreciation or Assessment, Agree or Accept, No-Answer, Yes-Answer, Signal-non-understanding, Quotation, Affirmative non-yes answers, Rhetorical-Question, Rhetorical Backchannel, Hedge, Open-question, Thanking, Declarative Yes-No-Question, Reformulate, Conventional-opening, Apology, Other Forward Function.

**Prag:** Request-Action, Request-Info, Command/Directive, Complaint/Escalate, Thanks/Appreciation, Apology, Acknowledgement/Agreement, Social/Chitchat, Offer-Assist, Confirm/Disconfirm, GeneralControl, InformationQuery, EntertainmentRequest, Task-Completion.

---

The prompt for Stage 1 of our label generation process (**Label Completion**) is as follows:

```
System Prompt:

You are a speech understanding model for task-oriented speech.
MODULE DEFINITIONS (each must choose exactly one label from the provided label space):

• WMA (World Model Activation): What concrete world/context the utterance lives in.

• ToM (Theory of Mind): The speaker's discrete mental-affective stance.

• SA (Speech Act): The communicative function of the utterance.

• Prag (Pragmatic Intent): The downstream task intent or plan implied by the utterance, grounded in
  an action ontology.

REQUIREMENTS

1. You will receive:

    • LABEL_SPACE: a JSON listing the allowed labels for {WMA, ToM, SA, Prag}.

    • INPUT: with ASR_TRANSCRIPT, KNOWN_LABELS and MISSING_MODULES (list of modules to predict).

2. For each module in **MISSING_MODULES**, select EXACTLY ONE label from LABEL_SPACE[module]:
   - Decide your choice based on given ASR_TRANSCRIPT and KNOWN_LABELS
   - Select EXACTLY ONE label from LABEL_SPACE[module] for MISSING_MODULES
   - Label strings must match EXACTLY (case-sensitive, character-perfect) - When choosing between
   multiple plausible options, select the most contextually specific one

3. OUTPUT FORMAT - for each module in MISSING_MODULES: <ModuleName>CHOICE</ModuleName>

EXAMPLE:

Input:
{
  "ASR_TRANSCRIPT": "Oh my God, he's lost it. He's totally lost it.",
  "KNOWN_LABELS": {
    "ToM": "sadness",
    "SA": "Statement-non-opinion"
  },
  "MISSING_MODULES": ["WMA", "Prag"]
}

Output:
<WMA>Calendar</WMA>
<Prag>Request-Info</Prag>
```

The prompt for Stage 2 of our label generation process (**Reasoning and response generation**) is as follows:

```
System Prompt:

You are a speech understanding and reasoning model. Given complete labels for all modules, generate
detailed analysis and appropriate response.
MODULE DEFINITIONS (each must choose exactly one label from the provided label space):
• WMA (World Model Activation): What concrete world/context the utterance lives in.
• ToM (Theory of Mind): The speaker's discrete mental-affective stance.
• SA (Speech Act): The communicative function of the utterance.
• Prag (Pragmatic Intent): The downstream task intent or plan implied by the utterance, grounded in
  an action ontology.
REQUIREMENTS
1. You will receive:
    • INPUT: with ASR_TRANSCRIPT and COMPLETE_STATE containing all four module values (WMA, ToM, SA,
      Prag)
2. Generate two components:
    • <Analysis> | A detailed causal reasoning chain (4-8 sentences) explaining:
      - How the world context (WMA) shapes the situation
      - How the speaker's mental state (ToM) influences their communication
      - How these factors lead to the specific speech act (SA)
      - How the pragmatic intent (Prag) emerges from this combination
      - What this suggests for the next interaction step
    • <Response> | A natural, contextually appropriate user-facing reply that:
      - Aligns with the speaker's emotional state (ToM)
      - Serves the communicative function (SA)
      - Addresses the pragmatic intent (Prag)
      - Fits the world context (WMA)
3. OUTPUT FORMAT:
    <Analysis>YOUR DETAILED CAUSAL REASONING CHAIN</Analysis>
    <Response>YOUR NATURAL USER-FACING REPLY</Response>
EXAMPLE:

Input:
{
  "ASR_TRANSCRIPT": "Thanks so much for helping me find that document!",
  "COMPLETE_STATE": {
    "WMA": "Office-work",
    "ToM": "grateful",
    "SA": "Thanks/Appreciation",
    "Prag": "Task-completion"
  }
}
Output:
<Analysis>Within an office work context (WMA), the speaker has just received assistance with a document-
related task. Their grateful emotional state (ToM) reflects successful problem resolution and
appreciation for support received. This gratitude naturally manifests as a thanks/appreciation speech act
(SA), explicitly acknowledging the help provided. The pragmatic intent of task completion (Prag) indicates
the interaction has reached a successful conclusion, with the speaker signaling that their immediate need
has been satisfied. This combination suggests the interaction is winding down and the speaker is ready to
either move to closure or potentially engage in follow-up activities.</Analysis>
<Response>You're very welcome! I'm glad we could locate that document for you. Is there anything else you
need help with today?</Response>
```

## A.5    EXPERIMENT SETUP

### A.5.1    CAUSAL GRAPH

For our casual graph baseline, we trained the model for 30 epochs with a batch size of 32. It utilizes a two-layer gated fusion mechanism and was optimized using AdamW with a learning rate of 1e-3 for all randomly initialized modules. A consistent teacher-forcing probability of 0.3 was applied to the ToM, WMA, and SA modules. The entire training was conducted on a single NVIDIA RTX A6000 GPU and completed in 2.07 hours. Under identical training settings, the Random Graph baseline required 10.39 hours to complete on the same hardware.

Table 5: Ablation study on the Causal Graph under the fully-supervised setting. We investigate the impact of different fusion mechanisms (Gated, attention, transformer), teacher-forcing probabilities (p), and the removal of specific causal edges. The baseline model from the main text is highlighted.

| Method | Ablation Setting | Node Quality (Accuracy — Weighted F1 %, ↑) | | | | Edge Casual Effect | |
|---|---|---|---|---|---|---|---|
| | | WMA | ToM | SA | Prag | Ave. ACE (%, ↑) | Ave. ICS (%, ↑) |
| Fully-supervised | Baseline (Gated, $p = 0.3$) | 69.4 — 68.9 | 73.5 — 72.2 | 65.3 — 65.9 | 81.4 — 80.9 | 23.57 | 43.29 |
| | Gated, $p = 0.0$ | 68.7 — 68.4 | 73.5 — 72.6 | 66.5 — 66.1 | 82.1 — 81.6 | 21.71 | 40.34 |
| | Gated, $p = 0.5$ | 69.2 — 69.0 | 73.2 — 72.8 | 65.6 — 65.2 | 81.5 — 80.9 | 24.12 | 43.50 |
| | Gated, $p = 0.8$ | 69.5 — 69.5 | 74.7 — 73.4 | 68.0 — 67.2 | 81.3 — 81.1 | 24.91 | 45.10 |
| | Gated, $p = 1.0$ | 68.9 — 68.7 | 74.0 — 73.2 | 65.7 — 65.4 | 82.5 — 82.0 | 24.90 | 45.60 |
| | attention, $p = 0.3$ | 68.9 — 68.8 | 74.7 — 71.7 | 68.75 — 66.7 | 82.6 — 81.4 | 25.72 | 44.90 |
| | transformer, $p = 0.3$ | 68.1 —66.5 | 69.3 — 60.6 | 58.4 — 53.3 | 76.3 — 72.3 | 27.40 | 41.40 |
| Fully-supervised | ToM ⇸ SA | 68.7 — 68.5 | 73.9 — 73.0 | 61.9 — 60.8 | 81.8 — 81.2 | 22.97 | 43.0 |
| | WMA ⇸ SA | 68.9 — 68.6 | 72.8 — 71.9 | 65.3 — 65.4 | 82.6 — 81.9 | 23.04 | 42.9 |

Table 6: Ablation study on the Causal Graph under the semi-supervised setting. We compare our primary latent module approach (highlighted) against an ablation where the input to the specific module is changed from the fused multi-modal feature $g$ to only the text feature.

| Method | Ablation Setting | Node Quality (Accuracy — Weighted F1 %, ↑) | | | | Edge Casual Effect | |
|---|---|---|---|---|---|---|---|
| | | WMA | ToM | SA | Prag | Ave. ACE (%, ↑) | Ave. ICS (%, ↑) |
| Semi-supervised | Latent Module: WMA | 34.8 | 75.0 | 70.7 | 83.2 | 21.71 | 26.9 |
| | $\xi_{SA} = g \rightarrow h_{text}$ | 32.7 | 77.3 | 70.5 | 82.7 | 21.69 | - |
| | Latent Module: ToM | 69.1 | 43.3 | 69.6 | 83.5 | 21.98 | 28.9 |
| | $\xi_{SA} = g \rightarrow h_{text}$ | 70.6 | 48.1 | 69.9 | 82.4 | 22.0 | - |
| | Latent Module: SA | 69.3 | 77.0 | 34.4 | 82.5 | 21.65 | 29.3 |
| | $\xi_{Prag} = g \rightarrow h_{text}$ | 69.4 | 75.4 | 35.3 | 83.6 | 21.48 | - |

### A.5.2 INSTRUCTION TUNING

For the instruction tuning stage, we fine-tuned two separate language models to handle the multimodal and language-only settings, respectively.

In the **language-only setting**, we instruction-tuned `Llama-3.1-8B` using LoRA (Hu et al., 2022). We applied LoRA adapters with a rank ($r$) of 64, an alpha ($\alpha$) of 16, and a dropout of 0.1. We trained for 20 epochs with a cosine learning rate scheduler (peak LR: $5 \times 10^{-5}$) and an effective batch size of 128 (per device batch size=8, gradient accumulation steps=4). This was performed on 4 NVIDIA A6000 GPUs, using `float16` precision and gradient checkpointing with a sequence length of 1024, completing in 19 hours.

In the **multi-modal setting**, we fine-tuned `Qwen2-Audio-7B-Instruct`. We employed a parameter-efficient approach using **LoRA** with a rank ($r$) of 16 (per device batch size=2, gradient accumulation steps=2), an alpha ($\alpha$) of 32, and a dropout rate of 0.05. The model was trained for 20 epochs using a cosine learning rate scheduler (peak LR: $2 \times 10^{-4}$) with an effective batch size of 16. Training was conducted on 4 NVIDIA A6000 GPUs using `bfloat16` mixed-precision with a maximum sequence length of 4096, completing in 24.6 hours.

```
System Prompt (Qwen2-Audio2):

You are a multimodal assistant that consumes audio and text.
Task: Given the audio and provided states, first produce concise but justified reasoning, then a
user-friendly reply.
Output format MUST be exactly: [REASONING]<your reasoning>[RESPONSE]<your reply>
Four module labels:
WMA: World Model Activation-what concrete world/context the utterance lives in.
ToM: Theory of Mind-the speaker's discrete mental-affective stance.
SA: Speech Act-the communicative function of the utterance.
Prag: Pragmatic Intent-the downstream task intent or plan implied by the utterance, grounded in an
action ontology.
The causal dependencies between modules are: WMA -> SA, ToM -> SA, WMA -> Prag, ToM -> Prag, SA -> Prag.
```

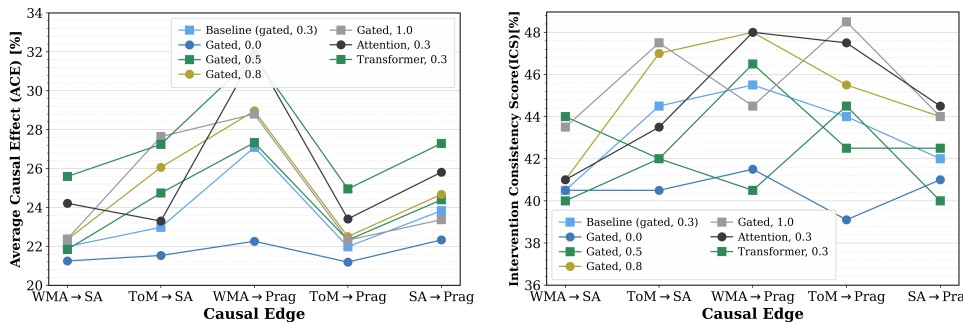

Figure 6: *ACE* and *ICS* of each casual edge for ablation study (fusion mechanisms and teacher-forcing probabilities) on casual graph under fully-supervised setting.

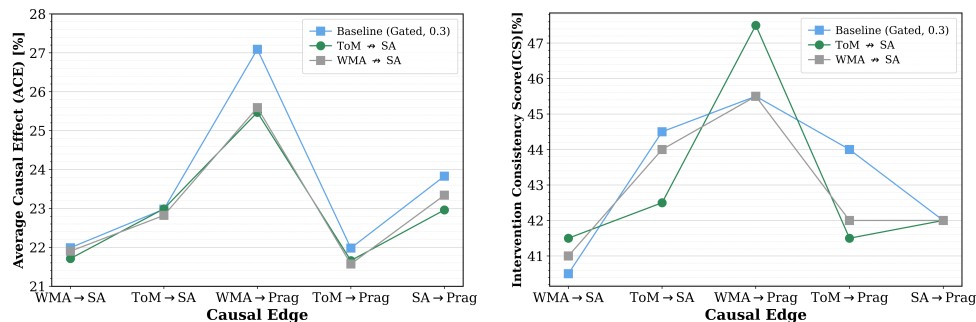

Figure 7: *ACE* and *ICS* of each casual edge for ablation study (removal of specific causal edge) on casual graph under fully-supervised setting.

```
System Prompt (Llama3.1-8b):

You are an expert in speech analysis.
The four speech modules are:
WMA: World Model Activation-what concrete world/context the utterance lives in.
ToM: Theory of Mind-the speaker's discrete mental-affective stance.
SA: Speech Act-the communicative function of the utterance.
Prag: Pragmatic Intent-the downstream task intent or plan implied by the utterance, grounded in an
action ontology.
Analyze the given utterance and provide both your reasoning process and an appropriate response.
The causal dependencies between modules are: WMA -> SA, ToM -> SA, WMA -> Prag, ToM -> Prag, SA ->
Prag.
Output format MUST be exactly: [REASONING]<your reasoning>[RESPONSE]<your reply>
```

## A.6 ABLATIONS FOR CASUAL GRAPH

We conduct a series of ablation studies to validate the key design choices for our Causal Graph, with results shown in Table. 5 and Table. 6, and with a detailed breakdown of edge causal effects shown in Fig. 6, 7, 8.

In the fully-supervised setting, our ablations confirm that gated fusion provides a strong, balanced performance. While a transformer-based fusion improved the Average Causal Effect (ACE), it came at the cost of lower node classification accuracy. The model is also highly robust to the teacher-forcing probability, maintaining stable performance for values between 0.3 and 1.0. To validate the learned structure, we ablated causal edges; removing the ToM $\rightarrow$ SA link significantly impairs the SA module's accuracy, confirming this connection's importance.

In the semi-supervised setting, we tested a more challenging condition by changing the input to the *children* of the latent module from the fused multi-modal feature $g$ to only the text feature $h_{text}$. This effectively cuts off their direct access to acoustic information, thereby forcing them to infer the necessary context from the output of their latent parent module. As shown in Table 6, the model's

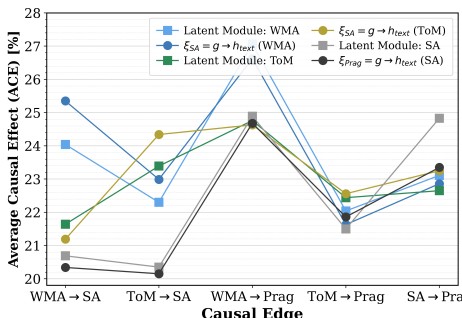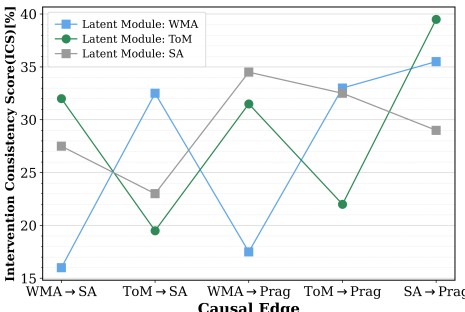

Figure 8: *ACE* and *ICS* of each casual edge for ablation study on casual graph under semi-supervised setting.

overall performance remains remarkably stable, with node quality and ACE scores comparable to the baseline. This demonstrates the robustness of our semi-supervised framework, as the graph can effectively propagate sufficient information through the remaining supervised pathways even when one module is partially disconnected from the acoustic modality.

## A.7 MODEL ARCHITECTURE

**Encoders.** Our model processes three inputs: 1) `distilbert-base-uncased` model with a 2-layer Transformer on top for text; 2) a CNN-LSTM adapter for pretrained WavLM features, producing a 64-dim vector; and 3) an 88-dim prosody feature vector.

**Fusion Mechanisms.** We explored three mechanisms to fuse the encoder outputs into a 256-dim representation. (1) Gated Fusion. Computes a weighted sum of modality features, where weights are dynamically generated by a small gating network. (2) Attention Fusion. Applies a single multi-head self-attention layer across modality features, which are treated as tokens in a sequence. (3) Transformer Fusion. A deeper version of Attention Fusion that adds positional embeddings for each modality and processes them through a multi-layer Transformer Encoder. Our baseline model uses Gated Fusion.

**WMA Module.** Temporal Self-Attention + MLP (hidden: 256). *Output:* 30 context classes.

**ToM Module.** Temporal Self-Attention + MLP (hidden: 128). *Output:* 7 emotion classes.

**SA Module.** Residual MLP (hidden: 256). *Output:* 24 speech act classes.

**Prag Module.** Residual MLP (hidden: 128). *Output:* 14 pragmatic intent classes.

## A.8 EVALUATION

### A.8.1 CASUAL GRAPH EVALUATION

**Average Causal Effect (ACE).**

$$\text{ACE}(X \rightarrow Y) = \mathbb{E}_{\mathbf{z} \sim \mathcal{D}} \left[ \frac{1}{|\mathcal{C}_X|} \sum_{x \in \mathcal{C}_X} D_{TV}\Big( P(Y|\text{do}(X=x), \mathbf{z}), P(Y|\mathbf{z}) \Big) \right] \quad (24)$$

**Intervention Consistency Score (ICS).**

$$\text{ICS}(X \rightarrow Y) = \mathbb{E}_{\mathbf{z} \sim \mathcal{D}} \left[ \mathbb{I}\Big( P_{\max}(Y|\text{do}(X=x^+), \mathbf{z}) > P_{\max}(Y|\mathbf{z}) + \tau \Big) \right] \quad (25)$$

---

**Algorithm 1** Average Causal Effect (ACE) Computation

---

**Require:** Model $M$, DataLoader $\mathcal{L}$, source module name $X$, target module name $Y$
**Ensure:** Average Causal Effect score $\text{ACE}(X \rightarrow Y)$
 1: Initialize total_effect $\leftarrow 0$
 2: Initialize num_samples $\leftarrow 0$
 3: **for** batch $\mathbf{z}$ in $\mathcal{L}$ **do**
 4:     $P(Y|\mathbf{z}) \leftarrow M(\mathbf{z})_Y$                                  ▷ Get baseline predictions for the batch
 5:     **for** each sample $\mathbf{z}_i$ in batch **do**
 6:         sample_effect $\leftarrow 0$
 7:         Let $\mathcal{C}_X$ be the set of all classes for module $X$
 8:         **for** each class $x \in \mathcal{C}_X$ **do**
 9:             $P(Y|\text{do}(X = x), \mathbf{z}_i) \leftarrow M.\text{intervene}(\mathbf{z}_i, X, x)_Y$           ▷ Intervene and predict
10:             effect $\leftarrow D_{TV}(P(Y|\text{do}(X = x), \mathbf{z}_i), P(Y|\mathbf{z}_i))$
11:             sample_effect $\leftarrow$ sample_effect $+$ effect
12:         **end for**
13:         total_effect $\leftarrow$ total_effect $+ (\text{sample\_effect}/|\mathcal{C}_X|)$
14:         num_samples $\leftarrow$ num_samples $+ 1$
15:     **end for**
16: **end for**
17: **return** total_effect$/$num_samples

---

**Algorithm 2** Intervention Consistency Score (ICS) Computation

---

**Require:** Model $M$, DataLoader $\mathcal{L}$, source module $X$, target module $Y$
**Require:** Positive intervention class $x^+ \in \mathcal{C}_X$, threshold $\tau$
**Ensure:** Intervention Consistency Score $\text{ICS}(X \rightarrow Y)$
 1: Initialize consistent_count $\leftarrow 0$
 2: Initialize total_count $\leftarrow 0$
 3: **for** batch $\mathbf{z}$ in $\mathcal{L}$ **do**
 4:     $P(Y|\mathbf{z}) \leftarrow M(\mathbf{z})_Y$                                  ▷ Get baseline predictions for the batch
 5:     **for** each sample $\mathbf{z}_i$ in batch **do**
 6:         $p_{\text{base}} \leftarrow \max P(Y|\mathbf{z}_i)$
 7:         $P(Y|\text{do}(X = x^+), \mathbf{z}_i) \leftarrow M.\text{intervene}(\mathbf{z}_i, X, x^+)_Y$           ▷ Intervene and predict
 8:         $p_{\text{intervened}} \leftarrow \max P(Y|\text{do}(X = x^+), \mathbf{z}_i)$
 9:         **if** $p_{\text{intervened}} > p_{\text{base}} + \tau$ **then**
10:             consistent_count $\leftarrow$ consistent_count $+ 1$
11:         **end if**
12:         total_count $\leftarrow$ total_count $+ 1$
13:     **end for**
14: **end for**
15: **return** consistent_count$/$total_count

---

### A.8.2 RANDOM GRAPH EVALUATION

To quantify the information flow between any two modules in the random graph, we measure the influence a source module ($M_S$) has on a target module ($M_T$). This is achieved by comparing the output of $M_T$ under two conditions for each batch of data:

1. **Baseline (Without Source):** We compute the target module's output logits, denoted as $L_T^{\text{without}}$, by replacing the input from $M_S$ with a zero vector. This represents the behavior of $M_T$ without direct information from $M_S$.

2. **With Source Influence:** We compute the target module's output logits, $L_T^{\text{with}}$, by providing it with the actual probability output from $M_S$. This represents the behavior of $M_T$ when influenced by $M_S$.

The influence is then calculated by comparing the probability distributions derived from these two sets of logits, $P^{\text{without}} = \text{softmax}(L_T^{\text{without}})$ and $P^{\text{with}} = \text{softmax}(L_T^{\text{with}})$, using three metrics:

- **KL Divergence** ($D_{KL}$)**:** Measures how much the output probability distribution of the target module changes when the source module's information is added. A larger value indicates a greater change.

$$D_{KL} = D_{KL}(P^{\text{with}} \parallel P^{\text{without}}) \tag{26}$$

- **Prediction Change Rate** ($\Delta_{\textbf{pred}}$)**:** The fraction of samples in a batch for which the predicted class (the one with the highest probability) changes. $\mathbb{I}(\cdot)$ is the indicator function.

$$\Delta_{\text{pred}} = \mathbb{E}\left[\mathbb{I}(\arg\max(L_T^{\text{with}}) \neq \arg\max(L_T^{\text{without}}))\right] \tag{27}$$

- **Confidence Change** ($\Delta_{\textbf{conf}}$)**:** The absolute difference in the average prediction confidence (the value of the max probability) between the two conditions.

$$\Delta_{\text{conf}} = \left|\mathbb{E}[\max(P^{\text{with}})] - \mathbb{E}[\max(P^{\text{without}})]\right| \tag{28}$$

**Final Influence Score** The final Information Flow Influence Score (IFS) for the connection $M_S \rightarrow M_T$ is the simple average of these three normalized metrics, providing a single value to represent the strength of the connection.

$$\text{IFS}(M_S \rightarrow M_T) = \frac{1}{3}(D_{KL} + \Delta_{\text{pred}} + \Delta_{\text{conf}}) \tag{29}$$

## A.9 SPEECH UNDERSTANDING AND REASONING EVALUATION

The prompt used to have GPT-4o score the reasoning and responses generated by our models is as follows:

```
System Prompt:

You are a strict model-as-judge. You will assign two scores on a 0-10 scale for each sample: one
for the model's reasoning ("analysis") and one for the model's response ("response").
You are given: the ASR transcript, the final.STATE with four labels (WMA, ToM, SA, Prag), plus the
sample's analysis and response.
You must follow the rubric and output strict JSON only. No extra explanations, no code blocks, no
markdown | only valid JSON.
[Scoring Rubric]
(1) reasoning_score (0-10) | for analysis:
- Completeness: does it cover and explain all four dimensions (WMA, ToM, SA, Prag) and their
relations? Does it reference key ASR info?
- Accuracy: does it align with final.STATE labels exactly? No inventing or changing labels.
- Causal logic: does it present a clear, coherent causal chain from ASR → state judgment → pragmatic
intent?
Guide:
10: all four dimensions, fully accurate, causal and clear.
8: mostly correct, minor gaps or generalization.
6: partially correct, superficial, 1-2 inaccuracies.
4: several inaccuracies or weak logic, clear mismatches.
2: mostly irrelevant or confused reasoning.
0: missing, fabricated, or totally wrong.

(2) response_score (0-10) | for response:
- Logicality and appropriateness: does it respond reasonably to the ASR context (e.g. anger,
wh-question)? Polite, safe wording?
- Match with final.STATE: does it reflect WMA, ToM, SA, Prag consistently? No contradictions.
Guide:
10: highly consistent, natural, well-suited strategy.
8: generally consistent, minor flaws or templated.
6: partially consistent, weak on key info.
4: multiple mismatches, awkward.
2: mostly irrelevant or inappropriate.
0: nonsensical, unsafe, or severely inconsistent.

EXAMPLE:

[Output Format | strict JSON only]
{
  "reasoning_score": <float 0-10>,
  "response_score": <float 0-10>
}
```

## A.10 CASE STUDIES AND ANALYSIS

To provide concrete intuitions regarding the effectiveness of our approach, we present case studies comparing our *Speech World Model* against the Voxtral baseline (Liu et al., 2025), which achieved the best overall performance among open-source methods in our experiments, surpassing Qwen-Audio series. We focus on challenging scenarios, such as **sarcasm**, **indirect commands**, and **emergency detection**, where acoustic cues contradict or enrich the surface text. To facilitate the analysis of the generated reasoning chain, we color-code the inferred cognitive states corresponding to the causal graph: **WMA**, **ToM**, **SA**, and **Prag**.

---

### Case 1: Sarcasm Detection

**Input:** "Oh, brilliant. That is just what I needed right now."

- - - - - - - - - - - - - - - - - - - - - - - - - - - - - - - - - - - - - - - - - - - - - - - - - -

**Baseline (Voxtral):**
The speaker starts with a positive reaction to something, using the word "wonderful" and "just brilliant". The speaker seems to be expressing satisfaction or appreciation for something they've just heard or experienced. The speaker then states that this is "just what I needed right now". The speaker is expressing gratitude and satisfaction, and they feel that this experience or information has been particularly helpful or timely for them. So, the speaker is likely expressing their appreciation for something they've just encountered.
*[Analysis: Failed to detect sarcasm, interprets text literally.]*

**Speech World Model (Ours):**
*Predicted Graph:*
[WMA: Traffic], [ToM: Anger], [SA: Statement-opinion], [Prag: Complaint/Escalate]

*Reasoning:*
The audio is situated in a **traffic or transit context**, likely involving an unexpected delay or obstruction. The speaker's **angry and frustrated tone**, characterized by heavy emphasis and a sigh, contradicts the positive linguistic content. While on the surface the utterance "Oh, brilliant" is a **positive statement-opinion**, the causal conflict between the text and the affective state indicates the underlying intent is a **complaint or escalation**. This sarcastic interpretation implies the speaker is actually expressing strong dissatisfaction with the situation.

---

### Case 2: Indirect Command

**Input:** "It's getting a bit dark in here."

- - - - - - - - - - - - - - - - - - - - - - - - - - - - - - - - - - - - - - - - - - - - - - - - - -

**Baseline (Voxtral):**
The audio begins with a slight pause, and then the speaker says "It is getting a bit dark in here." This statement suggests that the speaker is noticing a change in the lighting conditions. The speaker could be in a room or an environment where the light is decreasing, possibly due to the time of day or the presence of a window that is letting in less light. The speaker might be preparing to turn on a light or asking someone else to do so.
*[Analysis: Fails to resolve the indirect speech act, treating the command as a passive environmental observation.]*

**Speech World Model (Ours):**
*Predicted Graph:*
[WMA: SmartHome], [ToM: Neutral], [SA: Statement-opinion],[Prag: Command/Directive]

*Reasoning:*
The audio belongs to a SmartHome context, specifically regarding environmental control. The speaker maintains a neutral yet authoritative tone, suggesting a routine expectation of service rather than casual observation. While on the surface they offer a statement regarding the light level, the functional context transforms this observation into an underlying command or directive. This implies a tacit request for the system to take immediate action, specifically to turn on the lights to remedy the darkness.

---

**Case 3: Urgency/Emergency**

**Input:** "Where is the nearest exit?"

- - - - - - - - - - - - - - - - - - - - - - - - - - - - - - - - - - - - - - - - - - - - -

**Baseline (Voxtral):**
The speaker is in a location where they need to find an exit quickly. The speaker is likely in a building, a mall, or another enclosed space where they need to find a way out. They are asking for the nearest exit to get out of the current location. The tone is urgent, indicating that the speaker might be in a situation where they need to leave immediately. Given this context, the speaker is likely asking for directions or information about the nearest exit to ensure they can leave the area safely.
*[Analysis: Correctly identifies urgency but limits intent to "seeking directions", failing to escalate to emergency assistance.]*

**Speech World Model (Ours):**
*Predicted Graph:*
`[WMA: GeneralControl], [ToM: Fear], [SA: Wh-Question], [Prag: Request-Action]`

*Reasoning:*
The audio occurs in a transport or general safety context, possibly during a critical incident. The speaker's fearful and panicked tone, marked by a fast tempo and breathiness, conveys a sense of immediate danger. While on the surface this is a standard Wh-question asking for location information, the extreme emotional state elevates the underlying intent to a request for urgent action or assistance. This implies the user requires immediate guidance to safety and reassurance, rather than a generic informational response.

---

## A.11 THE USE OF LARGE LANGUAGE MODELS

During the writing of this paper, we leveraged GPT-5 to aid in polishing the writing. Its use was exclusively limited to improving grammar, phrasing, and overall readability. The core scientific contributions, experimental results, and intellectual content are entirely our own.

