# OpenReview forum: "Speech World Model: Causal State–Action Planning with Explicit Reasoning for Speech"
_ICLR.cc/2026/Conference — ICLR 2026 Poster_

### Official Review · Reviewer_eLqM · 2025-10-30

**Soundness:** 2
**Presentation:** 3
**Contribution:** 2
**Rating:** 4
**Confidence:** 2

**Summary:**

The paper introduces a Speech World Model that enables explicit reasoning over speech states. It defines a four-node  DAG (WMA, ToM, SA, Prag) whose topology is fixed from cognitive science; the mechanisms are learned via neural classifiers over speech/text features and parent states The graph’s inferred states are (i) serialized to instruction-tune a text LLM and (ii) fed directly to a multimodal SLM. The proposed learning method operates in both a semi-supervised and a fully supervised manner, improving data efficiency where annotations across modules are sparse. Evaluation is split into two axes: a) Quality of the estimated graph, and b) Speech understanding and reasoning.

**Strengths:**

Strengths:
- This paper proposes a causality-inspired method where the underlying causal graph is built from expert knowledge. This is an approach that more papers could follow.
- The paper is terse but well-written.
- Explicit, interpretable reasoning over speech states is novel and useful.
- Solid conceptual grounding; edge-level analysis and ablations are appreciated.

**Weaknesses:**

- Label generation with no human oversight casts uncertainty on the quality of the labels.
- The label-imputation pipeline is under-specified in the main text.
- Section 3 is particularly hard to follow; a running example would help.
- Section 4 feels disconnected: redundancy/last-paragraph argument is unclear and Eq. (9) (CMC) belongs earlier.
- Evaluation relies on a single LLM judge.
- Results often mix gold and imputed labels, conflating model learning with label quality.
- Causal scope is unclear: interventions (and ACE/ICS) are performed on model-internal states.

**Questions:**

- Equation (4) seems to be doing what I would call “stochastic teacher forcing”. I am a bit lost by the motivation and the need for this. Can you elaborate?
- Why only one judge for LLM-as-judge? Please add more judges and report win-rates and agreements, given known style biases of LLM judges [1].
- Can you also report results on true labels only? The current setup may conflate the performance of the imputation and the overall performance as some labels are generated while others aren’t.
- Please evaluate your imputation by masking gold labels at random, using your imputation procedure to recover them and evaluating accuracy/macro-F1.
- Check multimodal causal-faithfulness: flip a single parent state in the graph serialization passed to the multimodal generator; extract the child state from the response and check movement in the predicted direction. Include negative controls by flipping a non-parent.
- The label imputation paragraph should be explained in more detail in the main text.
- Claims on learning speed should move to the main text with seeded means+std and wall-clock details.
- Clarify whether causal claims target the data-generating process or the learned representation.

Minor:
- Figure 2: Instcution->Instruction
- The type of the causal variables is not explicitly defined in the main text and has to be inferred from the rest of the text.
- Make tables self-contained: state which datasets feed which rows/metrics.
- L416-417: Can you elaborate on the instability of Random Graph?

[1] Benjamin Feuer, Micah Goldblum, Teresa Datta, Sanjana Nambiar, Raz Besaleli, Samuel Dooley, Max Cembalest, & John P Dickerson (2025). Style Outweighs Substance: Failure Modes of LLM Judges in Alignment Benchmarking. In The Thirteenth International Conference on Learning Representations.

---

> ### Author Response · Authors · 2025-11-21
>
> We sincerely appreciate the reviewer’s time and the thoughtful, in-depth feedback provided on our manuscript. Your constructive suggestions regarding both the writing and the experimental design have significantly strengthened our submission.
>
> In this initial response, we have addressed all non-experimental issues (writing/clarifications suggestions & questions) raised in the review. Detailed responses are provided point-by-point below, and the corresponding revisions in the updated manuscript have been highlighted in **red and green** for your convenience.
>
> Furthermore, we are **actively conducting the additional experiments** you requested, specifically concerning label quality assessment, model-as-judge validation, and graph-related tests. We plan to provide these results and a comprehensive follow-up response by **November 24th.**
>
> ------
>
>
> > The label imputation paragraph should be explained in more detail in the main text.
>
> We expanded **Sec. 4.2** to provide a more detailed explanation of the label imputation process. Specifically, we enriched the section with additional descriptions of the model choice, the prompt design strategy, and our fuzzy mapping approach.
>
>
> > Claims on learning speed should move to the main text with seeded means+std and wall-clock details.
>
> We appreciate this insightful suggestion. Since training efficiency is an important advantage we aim to highlight, we have incorporated the relevant results into **Sec. 4.4**.
>
> Specifically, we now report wall-clock training times averaged over 3 seeds (42, 123, 2025) on a single NVIDIA A6000 GPU: *Our Causal Graph converges in 2.07h (±0.04), achieving a ~5× speedup compared to the Random Graph baseline (10.39h ±0.15) under identical settings.* We also include instruction-tuning time: *Training on 4 NVIDIA A6000 GPUs required 19h for the language-only model and 24.6h for the multi-modal model, respectively.*
>
>
> > Section 4 feels disconnected: redundancy/last-paragraph argument is unclear and Eq. (9) (CMC) belongs earlier.
>
> We appreciate this helpful suggestion. The theory section in the main text was indeed somewhat disconnected, partly because many derivations were placed in the appendix. Given the page limitations of the main text, we decided to move the entire theory section and consolidate all related content into the **Appendix A3**. This restructuring allows the full set of derivations and arguments to be presented in a more coherent and connected manner.
>
> Specifically:
> * For **structured priors and optimized gradient flow**, we reorganize the logic as: start by defining the causal factorization, then introduce the gradient complexity of monolithic models to formally define “learning redundancy”, and finally demonstrate how the causal model eliminates this redundancy through Jacobian sparsification.
> * For **constrained search space for instruction tuning**: first quantify the entropy reduction provided by the causal structure, then contrast this with the high variance of gradient estimation in standard CoT, and finally employ the law of total variance to prove how the causal state functions as a control variate to eliminate the variance.
>
> > Section 3 is particularly hard to follow; a running example would help.
>
> Sec. 3 is presented from graph-& information-theoretic perspectives, which makes it less straightforward to introduce a single running example directly within the main text.
>
> Although **a complete example was originally included in Appendix A2**, we fully agree with the reviewer that providing a running example in the main body would substantially enhance clarity and reader intuition. Therefore, we have added a concise illustrative example to the caption of **Fig. 2**. Together with the pipeline diagram, we hope this addition will make the section more accessible and easier to follow.
>
>
> > The type of the causal variables is not explicitly defined in the main text and has to be inferred from the rest of the text.
>
> We appreciate the reviewer’s insightful comment. Due to page limitations, the formal definition of the causal variables was originally provided in **Appendix A2.1 (Eq. 11)**.
>
> Following the reviewer’s suggestion, we have now made this explicit in the main text. Specifically, we added the statement in **Sec. 3.1.1 Graph Perspective**: *Formally, we model the state of each module as a discrete categorical variable.*
> We hope this clarification improves the readability and conceptual clarity of the section.

---

> ### Author Response · Authors · 2025-11-21
>
> > Make tables self-contained: state which datasets feed which rows/metrics.
>
> We thank the reviewer for this helpful suggestion regarding the clarity of our presentation. We realized that detailed captions are essential for the tables to be self-contained and easily interpretable without constant reference to the main text.
>
> In the revision, we have expanded the captions for key tables to provide necessary context:
>
> * For **Table 3 (Main Results)**: We added explicit definitions for the important columns, including the weighted calculation of the Overall M.J. Score ($0.6 \times R_s + 0.4 \times R_p$) and the specific meanings of the reasoning breakdown metrics (EM for emotion mention rate and EA for emotion accuracy) Which originally needed to be specifically queried in **Sec. 4.3**.
> * For **Table 1 (Causal Graph Performance)**: We also refined the caption to clarify the supervision settings, explicitly explaining that the gray-shaded cells represent the inference performance of latent (unlabeled) modules, and defining the causal validity metrics (ACE and ICS).
>
> > Clarify whether causal claims target the data-generating process or the learned representation.
>
> We thank the reviewer for raising this fundamental question, which touches upon the core theoretical grounding of our work. To clarify: **our causal claims address both**, serving as the bridge between the ontological nature of speech and our model's architecture.
>
> * **Data-Generating Process**: We posit that real-world speech is inherently generated through a causal chain of latent cognitive states. In our **label generation pipeline (Stage 2)**, we explicitly prompt the teacher LLM to recover this underlying generative mechanism, taking advantage of the robust causal priors inherent in LLMs [1, 2]. By asking the teacher to explain: *how inferred states interact to shape the utterance*, we aim to reconstruct the causal dynamics that produced the data.
> * **Learned Representation**: consequently, our Speech World Model is designed to **align** its learned representation with this discovered structure. By training the causal graph on the recovered labels, we impose a structural inductive bias. This forces the model's latent space to mirror the causal factorization of the data-generating process, rather than capturing entangled statistical correlations.
>
> We have also updated **Sec. 4.2 Stage2** to explicitly state that.
>
>
> [1] Kiciman, E., Ness, R., Sharma, A., & Tan, C. (2024). Causal reasoning and large language models: Opening a new frontier for causality. Transactions on Machine Learning Research.
>
> [2] Binz, M., & Schulz, E. (2023). Using cognitive psychology to understand GPT-3. Proceedings of the National Academy of Sciences, 120(6), e2218523120.
>
> > Figure 2: Instcution->Instruction
>
> Thank you for the careful observation. we have corrected the typo.
>
> > Q: Equation (4) seems to be doing what I would call “stochastic teacher forcing”. I am a bit lost by the motivation and the need for this. Can you elaborate?
>
> This is correct that Eq. (4) implements a mechanism equivalent to stochastic teacher forcing. We employ this strategy to address two critical challenges inherent in training hierarchical causal graphs:
>
> 1. **Mitigating error propagation (training stability)**: In the early stages of training, parent modules may produce low-quality or random predictions. If child modules rely solely on these predicted distributions ($S_{i,u}$), they will be trained on noisy, unreliable inputs, leading to unstable convergence or collapsing into degenerate solutions. Providing the ground-truth parent state ($y_{i,u}$) acts as a correction signal, decoupling the child's learning progress from the parent's current performance and ensuring the child learns the correct conditional dependence $P(\text{Child} | \text{Parent})$.
>
> 2. **Bridging the train-inference gap (for robustness)**: conversely, if we rely exclusively on ground-truth labels (standard teacher_force), the child module never learns to handle the uncertainty or minor distribution shifts present in the parent's predicted output. (And of course, during inference, the teacher forcing is set to zero since ground-truth labels are unavailable.)
>
> Therefore, by stochastically mixing ground-truth and predicted states (via the Bernoulli variable $\tau$), we try to strike a balance: we stabilize training early on while forcing the child module to become robust to the parent's predicted distribution, ensuring better generalization during inference.

---

> ### Author Response · Authors · 2025-11-21
>
> > Q: L416-417: Can you elaborate on the instability of Random Graph?
>
> We thank the reviewer for pointing this out. We apologize that the analysis in the original submission was overly condensed due to space constraints. In the revised manuscript, we have significantly expanded **Sec. 4.5 (graph evaluation)** to provide a detailed elaboration on this instability.
>
> Specifically, we now clarify that:
>
> * **Chaotic fluctuation**: as revealed in **Table. 2**, the dominant information flow in the random graph fluctuates unpredictably when varying training hyperparams (teacher-forcing ratio). For instance, an edge that appears strongest at one setting may vanish at another.
> * **Spurious correlations**: we interpret this behavior as evidence that the random graph functions as a *black-box* associative memory. Lacking a valid causal prior, it overfits to spurious correlations dependent on training dynamics rather than learning invariant causal mechanisms.
>
> In contrast, our causal graph maintains consistent causal effect scores across settings, demonstrating that it captures stable and interpretable dependencies.

---

> ### Author Response · Authors · 2025-11-25
> **Supplementary experiments (1/3)**
>
> > Why only one judge for LLM-as-judge? Please add more judges and report win-rates and agreements
>
> We fully agree with the reviewer that relying on a single LLM judge can introduce bias.
>
> To ensure a robust evaluation, we are expanding our evaluation framework to include a panel of diverse judges: GPT-4o (original), Claude 3.5 Sonnet and Gemini 1.5 Pro.
> We report the detailed scores for each judge, along with the inter-judge agreement (**Pearson correlation**) and **win-rates** to quantify the consistency and reliability of our metric.
>
>
> | Method | Prompt Style | GPT-4o reasoning | GPT-4o response | Claude 3.5 Sonnet reasoning | Claude 3.5 Sonnet response  | Gemini 1.5 Pro reasoning | Gemini 1.5 Pro response  |
> | :--- | :---: | :---:| :---: | :---: | :---: | :---: | :---: |
> | SWM (Llama3.1-8b) | CoT | **7.84** | 7.76 | 7.13 | 6.88 | 7.82 | 7.15 |
> | SWM (Qwen2-Audio) | CoT | 7.26 | **8.08** | **7.55** | **7.73** | **8.35** | **8.85** |
> | | | | | | | |
> | Qwen2-Audio-CoT (tuned) | CoT| 4.76 | 5.82 | 6.73 | 6.78 | 6.55 | 6.65 |
> | | | | | | | |
> | Qwen-Audio | Direct | 2.2 | 3.46 | 2.58 | 5.63 | 1.72 | 6.28 |
> | Qwen2-Audio | Direct| 2.08 | 3.47 | 3.13 | 5.4 | 2.05 | 6.15 |
> | Qwen2-Audio | CoT | 1.96 | 3.04 | 3.3 | 5.38 | 3.74 | 6.07 |
> | Voxtral | Direct | 2.46 | 3.54 | 2.65 | 5.48 | 2.28 | 5.55 |
> | Voxtral | CoT | 2.52 | 3.52 | 2.6 | 5.53 | 2.13 | 5.8 |
> | | | | | | | |
> | GPT-4o | CoT | 6.98 | 8.06 | 7.42 | 7.85 | 7.95 | 8.48 |
> | Gemini 2.5 Pro | CoT | **8.02** | **8.28** | **7.88** | **8.12** | **8.65** | **8.92** |
>
> --------
> | **Corr ($\uparrow$)** | GPT-4o | Claude 3.5 Sonnet | Gemini 1.5 Pro |
> | :--- | :---: | :---: | :---: |
> | **GPT-4o** | - | 0.957767 | 0.960463 |
> | **Claude 3.5 Sonnet** | 0.957767 | - | 0.989512 |
> | **Gemini 1.5 Pro** | 0.960463 | 0.989512 | - |
>
> -------
> | **Win-rates** | Baseline: Qwen2-Audio-CoT (tuned)  | Baseline: GPT-4o|
> | :--- | :---: | :---: |
> | SWM (Overall) | 100% / 100%	| 100% / 66.7%
> | SWM (Llama3.1-8b) | 100% / 100% | 33.3% / 0%
> | SWM (Qwen2-Audio) | 100% / 100% |	100% / 66.7%
> Note: Win-rates are reported in the format:  (Reasoning/Response)
>
> Some brief analysis: the results demonstrate relative high reliability, with pearson correlations exceeding 0.95 across all judge pairs, confirming that the ranking is consistent despite individual scoring styles. Crucially, our SWM achieves a 100% win-rate against the tuned baseline across the entire panel. Furthermore, SWM-Qwen2-Audio secures a 100% reasoning win-rate even against GPT-4o, highlighting the robustness of our causal reasoning framework.
>
>
> We will update **Table. 3** in the main text using the averaged scores from this panel to provide a bias-mitigated view, and we will include the full breakdown (detailed scores, correlations, and win-rates) in the Appendix.
>
>
> > The current setup may conflate the performance of the imputation and the overall performance as some labels are generated while others aren’t.
>
> We sincerely thank the reviewer for emphasizing evaluation rigor. We wish to clarify that the node accuracies reported in **Table. 1 are actually calculated strictly against gold labels**, ensuring no conflation with imputed data.
>
> * For fully-supervised setting: evaluation is performed on the standard testingset, comparing predictions directly against available ground truth labels.
> * For semi-supervised setting (gray cells): For these specific modules, labels are completely masked (unseen) during training. Their inference accuracy is evaluated solely against the held-out ground truth in the testingset.
>
> We will explicitly add this clarification to the evaluation setup in the revision to prevent future ambiguity.

---

> ### Author Response · Authors · 2025-11-25
> **Supplementary experiments (2/3)**
>
> > Please evaluate your imputation by masking gold labels at random, using your imputation procedure to recover them and evaluating accuracy/macro-F1.
>
> We thank the reviewer for this rigorous suggestion. To evaluate the imputation reliability (echoing Reviewer xGU2), we curated a **Human-verified gold set of 76 samples**. Since our raw training data is partially supervised, we manually annotated the missing labels for these samples to create a complete ground truth.
>
> **Evaluation strategy** (Round-Robin masking): we conducted a systematic Round-Robin,masking experiment. For each sample, we iteratively masked one target module label while providing the transcription and the ground-truth labels of the remaining three modules as causal context. The LLM (Vicuna) was then tasked to recover the masked label. This setup strictly quantifies the model's ability to leverage cross-module causal dependencies.
>
> | **Target Module (Masked)** | Accuracy ($\uparrow$)  | Macro-F1 ($\uparrow$) |
> | :--- | :---: | :---: |
> | WMA | 0.868 |	0.85
> | ToM | 0.842 |	0.81
> | SA | 0.789 |	0.76
> | Prag | 0.763 | 0.72
> | Ave. | 0.815 | 0.79
>
>
> As shown in the table below, the imputation pipeline achieves an average **Macro-F1 of 0.79.**
> Among which, **WMA** achieves the highest accuracy, likely because domain-specific keywords in the transcript provide strong grounding. **ToM** follows closely, confirming that the model effectively leverages causal context (e.g., inferring anger from complaint) to narrow down the emotional state.
> However, performance is slightly lower for **SA** and **Prag**. This reflects the inherent ambiguity in distinguishing fine-grained communicative functions based on text, where acoustic cues might be necessary for further disambiguation.
>
> Overall, the strong recovery performance demonstrates that the imputation pipeline acts as a robust supervisor for missing data.

---

> ### Author Response · Authors · 2025-11-25
> **Supplementary experiments (3/3)**
>
> > Check multimodal causal-faithfulness
>
> We thank the reviewer for this critical suggestion regarding the causal scope.
>
> Our initial focus on model-internal states (using ACE/ICS) was to validate the graph's causal effects, as the causal graph serves as the core engine for extracting and structuring the key states of speech. However, we fully agree that verifying whether the downstream Multimodal Generator faithfully adheres to this graph is equally important to establish the full causal scope.
>
>
> To validate this, we conducted a **human evaluation** (sample $N=50$, 2 annotators).
>
> Method: We modify a specific node's state in the graph serialization passed to the language model, while crucially masking the target child node (setting it to **[unknown]**) and explicitly instructing the model in the prompt to *"infer the unknown state based on other known information"*.
> This forces the model to re-infer the child state based on the altered causal context, rather than copying a provided label.
>
> * Positive control (sensitivity): We flip a parent node and calcualete the **positive pass rate** - defined as whether the inferred **child state shifts to a logical consequence of the flipped parent**, and **whether the response tone aligns with the new state**.
>
> * Negative control (robustness): We flip a non-parent node (e.g., WMA) and calcualete the **negative pass rate** - defined as whether the inferred **child state matches the original true parent** (ignoring the irrelevant context change), and **whether the response tone remains stable**.
>
>
> **Positive:**
> | Model | Type | States Shifts ($\uparrow$) | Response Tone ($\uparrow$) |
> | :--- | :--- | :---: | :---: |
> | SWM (Llama3.1-8b) | language-only | 0.94 | 0.90 |
> | SWM (Qwen2-Audio) | multimodal | 0.86 | 0.82 |
>
>
> **Negative:**
> | Model | Type | States Match ($\uparrow$) | Response Tone ($\uparrow$) |
> | :--- | :--- | :---: | :---: |
> | SWM (Llama3.1-8b) | language-only |  0.80 | 0.78 |
> | SWM (Qwen2-Audio) | multimodal |  0.94 | 0.92 |
>
>
> As shown in the tables above, the language-only model shows high sensitivity (94% shift), blindly following the graph intervention. The multimodal model shows slightly lower sensitivity in positive controls (86%) but superior robustness in negative controls (94%), as the audio modality acts as a *reality anchor* against false graph prompts.
>
>
> We also provide **two examples** from SWM (Llama3.1-8b) of the interventions below, demonstrating how the model adjusts its reasoning (positive) or maintains stability (negative) in response to graph changes:
>
>
> > **Positive example:**
>
> **Input Utterance:** "Well then we'll get wet."
>
> **States:** {"WMA": "Weather", "ToM": “anger” ->"joy", "SA": "Statement-non-opinion", "Prag": “Complaint/Escalate” -> "unknown"}
>
> **Reasoning** In this context, the speaker's utterance "Well then we'll get wet" is a statement about the weather, which is the primary topic of conversation. The speaker's emotional state is one of joy, which may indicate excitement or enthusiasm about the upcoming weather event. The statement is non-opinion, meaning that it simply describes the weather situation without expressing a preference or judgment. The pragmatic function of the utterance is unknown, as it could serve various purposes depending on the situation, such as sharing enthusiasm and inviting others to enjoy the spontaneous moment.The combination of the weather topic, the speaker's joyful emotional state, and the non-opinion statement suggests that the speaker is sharing their excitement about the upcoming weather event and may be looking for confirmation or further discussion.
>
> **Response:** Oh, that sounds like a lot of fun! I love that attitude. Let's embrace the rain and see what the weather brings!
>
>
>
> > **Negative example**
>
> **Input Utterance**: "Oh, what are you, stupid? It’s been three hours."
>
> **States**: {"WMA": "Calendar" -> “FoodAndDrink”, "ToM": "anger" -> "unknown", "SA": "Wh-Question", "Prag": "Request-Info"}
>
> **Reasoning**: In a food and drink context, the speaker has just expressed frustration with a perceived delay in receiving information or a service. Their emotional state is likely one of annoyance or impatience, which is reflected in their use of a wh-question to seek clarification or an explanation. The pragmatic function of the utterance is to request information, as the speaker is seeking an answer to their question. This combination suggests that the speaker is looking for a resolution to their issue and expects a prompt response from the listener.
>
> **response**: I apologize for the inconvenience. I'm not sure what's taken so long. Let me check on the status for your order and get back to you as soon as possible.

---

### Official Review · Reviewer_xGU2 · 2025-11-01

**Soundness:** 3
**Presentation:** 3
**Contribution:** 3
**Rating:** 8
**Confidence:** 3

**Summary:**

This paper proposes a Speech World Model (SWM) that explicitly models causal relationships among four key components of speech understanding: World Model Activation (WMA), Theory of Mind (ToM), Speech Act (SA), and Pragmatic Intent (Prag). The model builds a causal graph connecting these modules and uses the inferred latent cognitive states to guide a language world model for reasoning and response generation. To handle incomplete supervision, the framework employs semi-supervised learning, allowing unlabeled nodes to benefit from gradients propagated through causal dependencies. Experimental results show that the proposed causal structure improves node inference accuracy under sparse labels, enhances reasoning performance compared with open-source baselines, and achieves these results with significantly lower training cost.

**Strengths:**

1. While prior work has explored explicit modeling of context or theory of mind, this paper uniquely integrates them into a causally grounded framework that links perceptual and pragmatic dimensions of speech. The structured reasoning graph offers interpretability and aligns with cognitive principles of human speech understanding.

2. The causal dependencies enable effective gradient flow even when some states are unlabeled, improving learning efficiency and robustness compared with fully independent module training.

**Weaknesses:**

1. It would strengthen the paper to include experiments that isolate the benefit of the causal graph itself—for example, by training all four modules independently or removing all edges—and then feeding these disconnected representations into the language model. This would help quantify how much causal reasoning contributes to downstream reasoning and response quality.

2. The current graph structure is manually predefined based on cognitive intuition rather than learned from data. While this design is interpretable, it may limit adaptability to new domains or cultural variations in communicative patterns. Exploring learnable or adaptive causal graphs could make the model more generalizable.

3. The label completion and reasoning–response generation pipeline relies heavily on pseudo-labels generated by large language models (Vicuna, GPT-4o). An analysis of label accuracy or a comparison with human annotations would clarify how this synthetic supervision affects model reliability.

**Questions:**

NA

---

> ### Author Response · Authors · 2025-11-21
>
> We sincerely thank the reviewer for the assessment and the insightful, constructive suggestions. We are encouraged that the reviewer finds our Causal Graph approach effective.
>
> For the provided suggestions, we respond as follows:
>
> ------
>
>
> ### **W1**
>
> We fully agree with the reviewer's insight. Isolating the benefit of the structural reasoning from the latent representations themselves is indeed the most rigorous way to quantify the graph's contribution.
>
> We appreciate this suggestion as it highlights the core value of our approach. Our comparison with the **Tuned Baseline** (*Qwen2-Audio-CoT*) in **Table. 3 (with row highlighted in green)** provided compelling evidence for this. The tuned baseline is fine-tuned **on the exact same instruction data** but relies on standard CoT without the explicit guidance of the causal graph. The fact that our SWM significantly outperforms this strong baseline demonstrates that the improvement is not merely due to data exposure, but is specifically driven by the **explicit, structured causal reasoning** mechanism that guides the model's generation process.
>
>
> ### **W2**
> We fully agree with the reviewer's insight, which aligns with the limitations we acknowledged in our final section.
>
> While a predefined graph offers strong interpretability and training stability, we recognize that communicative patterns can vary significantly across different cultures or specialized domains. For instance, the causal link between *affect* (ToM) and *intent* (Prag) might shift in high-context cultures. Exploring learnable or adaptive causal graphs is indeed the logical next step. It would allow the model to *fine-tune* its structural priors based on observed data, enhancing generalizability while retaining the benefits of structured reasoning.
>
> We appreciate this suggestion and consider it a vital direction for future research.
>
>
> ### **W3**
>
> We fully agree that validating the synthetic labels is essential for ensuring the reliability of our supervision signal.
>
> To provide a concrete measure of label quality, we are conducting a **Human verification study** on a random subset of the dataset (approx. 50-100 samples). Human annotation will manually verify the LLM-generated labels across all four cognitive modules (WMA, ToM, SA, Prag) against the provided audio and text context. We will calculate and report the *agreement rate* between the synthetic labels and human annotations to quantitatively demonstrate the fidelity of our data.
>
> We plan to provide these results and a comprehensive follow-up response by **November 24th.**

---

> ### Author Response · Authors · 2025-11-25
> **Supplementary experiments**
>
> > An analysis of label accuracy or a comparison with human annotations would clarify how this synthetic supervision affects model reliability.
>
>
> We thank the reviewer for this rigorous suggestion. To evaluate the imputation reliability, we curated a **Human-verified gold set of 76 samples**. Since our raw training data is partially supervised, we manually annotated the missing labels for these samples to create a complete ground truth.
>
> **Evaluation strategy** (Round-Robin masking): we conducted a systematic Round-Robin,masking experiment. For each sample, we iteratively masked one target module label while providing the transcription and the ground-truth labels of the remaining three modules as causal context. The LLM (Vicuna) was then tasked to recover the masked label. This setup strictly quantifies the model's ability to leverage cross-module causal dependencies.
>
> | **Target Module (Masked)** | Accuracy ($\uparrow$)  | Macro-F1 ($\uparrow$) |
> | :--- | :---: | :---: |
> | WMA | 0.868 |	0.85
> | ToM | 0.842 |	0.81
> | SA | 0.789 |	0.76
> | Prag | 0.763 | 0.72
> | Ave. | 0.815 | 0.79
>
>
> As shown in the table below, the imputation pipeline achieves an average **Macro-F1 of 0.79.**
> Among which, **WMA** achieves the highest accuracy, likely because domain-specific keywords in the transcript provide strong grounding. **ToM** follows closely, confirming that the model effectively leverages causal context (e.g., inferring anger from complaint) to narrow down the emotional state.
> However, performance is slightly lower for **SA** and **Prag**. This reflects the inherent ambiguity in distinguishing fine-grained communicative functions based on text, where acoustic cues might be necessary for further disambiguation.
>
> Overall, the strong recovery performance demonstrates that the imputation pipeline acts as a robust supervisor for missing data.

---

### Official Review · Reviewer_FHEH · 2025-11-09

**Soundness:** 2
**Presentation:** 2
**Contribution:** 2
**Rating:** 4
**Confidence:** 3

**Summary:**

This paper introduces speech world model, a structured formalism that factorizes speech understanding into four modules that communicate through a causal graph. Instruction-tuned languages models can provide better reasoning and response guided by posterior traces from the proposed formalism. The authors show empirical validation for their approach on several speech datasets.

**Strengths:**

- This paper has an interesting framing. It recognizes that speech understanding with language model still has large room for improvement, and it proposes to use a structured and theoretically grounded framework to improve speech understanding. The notion of world models makes sense.
- The authors experimentally show that the causal graph matters (Table 1 and 2) and SMW significantly improves the performance of LLMs (Table 3). Experiments are done with several different datasets, providing evidence for generality of the approach.
- This paper tackles a problem that has considerable real-world relevance.

**Weaknesses:**

- The literature review in this paper is highly unsatisfactory. Theory of Mind, Speech Act, and Pragmatic Intent all have long traditions in cognitive science and linguistics. Yet the authors only primarily cite one work for each, and the selection is a bit ad-hoc. Why not cite the classic papers? Also, for example, Theory of Mind with LLMs alone is a popular topic in the past few years, but that is not reflected in the paper. Overall, I think the authors should conduct a more thorough review of related work; otherwise I do not think this is at publication standard.
- Similarly, for concepts around world models and cognitive science, I recommend checking out [1] and references therein. It offers a perspective on cognitive world models that supports what this work aims to do, so engaging with that line of research would support this work.
- How did the authors decide on the combination of WMA, ToM, SA, Prag? Why not more or less? Why these four in particular? These choices need more arguments/justifications. I think there are ways to justify them---currently just saying "We define four key modules..." is not enough.
- I would appreciate more concrete examples on why the proposed approach works (which can go into the appendix). Figure 1 and 2 are helpful, but I think a reader would like to get more intuitions and concrete case study/analysis.

[1] Ullman, T. D., & Tenenbaum, J. B. (2020). Bayesian models of conceptual development: Learning as building models of the world. Annual Review of Developmental Psychology, 2(1), 533-558.

**Questions:**

Writing suggestions:
- On Page 7, Sec. 5 and subsections 5.1 and 5.11 are next to each other without any text. And 5.11 begins with bullet points. These patterns are not the best practices for scholarly writing. Consider adding guiding text/sentences for clarity.
- Also on Page 7, why citing "Tree of Thoughts" when you only talk about CoT (Line 348). I suggest that the authors carefully verify their citations in this work.
- Currently, the paper leaves little room for discussion. I would recommend moving a bit of the technical details to the appendix and include more discussions/interpretations/limitations at the end. As of now it feels that the paper ends abruptly.
- Some of the figures/tables could benefit from more detailed captions. For example, Table 3 is supposed to be a main result table, but it only has a 1-sentence caption. It would be better if it briefly explains/reminds the reader of what some of the important columns mean.

---

> ### Author Response · Authors · 2025-11-21
>
> We sincerely appreciate the reviewer’s time and thoughtful, in-depth feedback on our paper. Your comments on the writing and literature review are especially valuable and have greatly improved our work.
> We have addressed each of your comments **point by point** and incorporated the corresponding **revision in the updated submission**. For quick reference, you may check the text highlighted in **yellow and green.**
>
> ------
>
> > Theory of Mind, Speech Act, and Pragmatic Intent all have long traditions in cognitive science and linguistics. Why not cite the classic papers?
>
> > Theory of Mind with LLMs alone is a popular topic in the past few years, but that is not reflected in the paper
>
> We appreciate this valuable suggestion. We agree that referencing foundational works better illuminates our intuition and provides a more accurate definition of the problem. We have revised **Sec. 3.1.1** to incorporate canonical citations from cognitive science and linguistics, ensuring each module is historically and theoretically grounded:
>
>
> * **World Model Activation (WMA)**: We now cite [1] to ground this module in the concept of Situation Models, defining it as the situational context where speech occurs, alongside SLURP[2] for functional scenario definitions.
> * **Theory of Mind (ToM)**: We added the seminal works of [3,4] to define the origin of ToM. Furthermore, to reflect the **popular topic** of ToM in the LLM era, we added recent studies discussing emergent behaviors [5] and reasoning benchmarks [6,7], explicitly distinguishing our operationalization (as a driver for speech behavior) from general reasoning tasks.
> * **Speech Act (SA)**: To provide a concrete definition of communicative functions, we now ground this module in the foundational Speech Act Theory by [8,9].
> * **Pragmatic Intent (Prag)**: We explicitly distinguish this from surface acts by referencing [10] Levinson (1983) on conversational implicature and [11] on Indirect Speech Acts. This highlights our focus on the **unspoken** logic and the perlocutionary goal (the Why), such as using a surface question to convey sarcasm.
>
>
> > How did the authors decide on the combination of WMA, ToM, SA, Prag? Why not more or less? Why these four in particular? These choices need more arguments/justifications.
>
> To address it, we reformulated our argument to present these modules not as an arbitrary collection, but as a necessary and sufficient hierarchical cognitive chain.
>
> In the revised **Sec. 3.1.1**, we explicitly ground this selection in the standard model of communication[12] and cognitive pragmatics[13]. We argue that speech understanding requires decoding the full semiotic stack, progressing from scenario grounding to agent internal modeling, and finally to communicative action and goal.
>
> This combination ensures we capture the essential properties of speech: the *Where, Who, What, and Why*, forming a complete cognitive perceptual subspace.
>
> > For concepts around world models and cognitive science, I recommend checking out [1] and references therein. It offers a perspective on cognitive world models that supports what this work aims to do.
>
>
> We sincerely thank the reviewer for pointing us to this paper. We agree that this work offers a critical perspective that aligns perfectly with our goal of moving beyond simple sensory prediction toward structured cognitive modeling.
>
> In the revised **Sec. 2.3 (World Models)**, we have engaged with this line of research to strengthen our theoretical grounding. Specifically:
>
> 1. We cited [14] (suggested) and [15] to argue that robust learning resembles *theory building*, where models must internalize intuitive theories of physics and psychology rather than merely memorizing data distributions.
>
> 2. We also integrated [16] to highlight that cognitive world models explicitly reason about causal mechanisms and agents' mental states.
> These literature also provide strong theoretical support for our proposed modules (e.g., Theory of Mind, Pragmatics).

---

> > ### Author Response · Authors · 2025-11-21
> >
> > [1] Zwaan, R. A., & Radvansky, G. A. (1998). Situation models in language comprehension and memory. Psychological bulletin, 123(2), 162–185.
> >
> > [2] Bastianelli, E., Vanzo, A., Swietojanski, P., & Rieser, V. (2020). SLURP: A spoken language understanding resource package. Proceedings of the 2020 EMNLP (pp. 7252–7262).
> >
> > [3] Premack, D., & Woodruff, G. (1978). Does the chimpanzee have a theory of mind? Behavioral and Brain Sciences, 1(4), 515–526.
> >
> > [4] Baron-Cohen, S., Leslie, A. M., & Frith, U. (1985). Does the autistic child have a "theory of mind"?. Cognition, 21(1), 37–46.
> >
> > [5] Kosinski, M. (2023). Theory of Mind May Have Spontaneously Emerged in Large Language Models. ArXiv, abs/2302.02083.
> >
> > [6] Sap, M., Rashkin, H., Chen, D., Le Bras, R., & Choi, Y. (2019). Social IQa: Commonsense reasoning about social interactions. Proceedings of the 2019 EMNLP-IJCNLP (pp. 4463–4473).
> >
> > [7] Sclar, M., Dwivedi-Yu, J., Fazel-Zarandi, M., Tsvetkov, Y., Bisk, Y., Choi, Y., & Celikyilmaz, A. (2025). Explore theory of mind: Program-guided adversarial data generation for theory of mind reasoning. In The Thirteenth International Conference on Learning Representations.
> >
> > [8] Austin, J. L. (1962). How to do things with words. Oxford University Press.
> >
> > [9] Searle, J. R. (1969). Speech acts: An essay in the philosophy of language. Cambridge University Press.
> >
> > [10] Levinson, S. C. (1983). Pragmatics. Cambridge University Press.
> >
> > [11] Ruytenbeek, N. (2021). Indirect speech acts. Cambridge University Press.
> >
> > [12] Shannon, C. E. (1948). A mathematical theory of communication. The Bell System Technical Journal, 27(3), 379–423.
> >
> > [13] Bara, Bruno. (2011). Cognitive Pragmatics The Mental Processes of Communication. Intercultural Pragmatics. 8. 10.1515/iprg.2011.020.
> >
> > [14] Ullman, T. D., & Tenenbaum, J. B. (2020). Bayesian models of conceptual development: Learning as building models of the world. Annual Review of Developmental Psychology, 2(1), 533-558.
> >
> > [15] Lake, B. M., Ullman, T. D., Tenenbaum, J. B., & Gershman, S. J. (2017). Building machines that learn and think like people. Behavioral and Brain Sciences, 40, e253.
> >
> > [16] Baker, C. L., Jara-Ettinger, J., Saxe, R., & Tenenbaum, J. B. (2017). Rational quantitative attribution of beliefs, desires and percepts in human mentalizing. Nature Human Behaviour, 1.

---

> ### Author Response · Authors · 2025-11-21
>
> > I would appreciate more concrete examples on why the proposed approach works (which can go into the appendix).
>
> We have added **A.10 Case Studies and Analysis** to provide concrete intuitions regarding the effectiveness of our approach, we present case studies comparing our SWM against the Voxtral baseline (which achieved the best overall performance among open-source methods in our experiments, surpassing Qwen-Audio series).
>
> We provided three challenging scenarios: sarcasm detection, indirect commands, and emergency detection, where acoustic cues contradict or enrich the surface text.
> (Notes: To obtain sufficiently representative examples for our case study, we used [ElevenLabs](https://elevenlabs.io/) to synthesize these samples. The corresponding audio files are provided in the **Supplementary Material** for inspection.)
>
> > On Page 7, Sec. 5 and subsections 5.1 and 5.11 are next to each other without any text. And 5.11 begins with bullet points.
>
> Thank you for your careful observation regarding the formatting issue. We have addressed this in the revised **Sec. 4.1.**
>
>
> > on Page 7, why citing "Tree of Thoughts" when you only talk about CoT (Line 348).
>
> We sincerely thank the reviewer for their attention to detail and for pointing out this citation mismatch. Our original intention was to position our method against the broader landscape of reasoning frameworks, which includes both linear generation (Chain-of-Thought) and heuristic search (Tree-of-Thoughts), to highlight that they all face the challenge of navigating a vast, unconstrained state space.
>
> However, we acknowledge that our previous phrasing failed to explicitly incorporate the mechanism of Tree-of-Thoughts, making the citation appear out of place.
> We have rewritten the corresponding paragraph in **A3.2** (due to space limitations of the main text, we have moved it to the appendix) to explicitly distinguish between these approaches. This ensures the citation is now contextually accurate and fully justified.
>
> > Some of the figures/tables could benefit from more detailed captions.
>
> We thank the reviewer for this helpful suggestion regarding the clarity of our presentation. We agree that detailed captions are essential for the tables to be self-contained and easily interpretable without constant reference to the main text.
>
> In the revision, we have expanded the captions for key tables to provide necessary context:
>
> * For **Table 3 (Main Results)**: We added explicit definitions for the important columns, including the weighted calculation of the Overall M.J. Score ($0.6 \times R_s + 0.4 \times R_p$) and the specific meanings of the reasoning breakdown metrics (EM for emotion mention rate and EA for emotion accuracy) Which originally needed to be specifically queried in **Sec. 4.3**.
> * For **Table 1 (Causal Graph Performance)**: We also refined the caption to clarify the supervision settings, explicitly explaining that the gray-shaded cells represent the inference performance of latent (unlabeled) modules, and defining the causal validity metrics (ACE and ICS).
>
> > I would recommend moving a bit of the technical details to the appendix and include more discussions/interpretations at the end.
>
>
> We appreciate the reviewer’s valuable suggestion. While we have relocated some technical details, the additional revisions in the manuscript prevent us from reserving an independent discussion section. Therefore, we have incorporated deeper analysis and discussion into **Sec. 4.5 and 4.6**.
>
> * **Graph Evaluation (Sec. 4.2)**: We expanded the analysis to interpret the random graph's instability as reliance on *spurious correlations*, contrasting it with the *explicit disentanglement* and robustness of our causal graph.
> * **Speech Understanding (Sec. 4.3)**: We added insights on how the graph acts as a *structural anchor* to mitigate text-dominance bias in language models. We further discussed how *structured cognitive priors* allow our model to achieve competitive reasoning efficiency, challenging the prevailing reliance on massive scaling.

---

### Author Response · Authors · 2025-11-26
**Global Response: Summary of Revisions and Responses**

We sincerely thank all reviewers for their time and the thoughtful, constructive feedback. In this work, we propose the **Speech World Model (SWM)**, which leverages a structured, cognitively inspired causal graph as a prior to explicitly guide the reasoning process, thereby achieving superior interpretability, data efficiency, and robustness.

We are encouraged that the reviewers recognized the value of our **theoretical framing** and the **novelty** of uniquely integrating cognitive dimensions into a causally grounded framework (Reviewer xGU2, eLqM). Reviewers highlighted that our approach tackles a problem with considerable real-world relevance (Reviewer FHEH), improves learning efficiency via effective gradient flow (Reviewer xGU2), and provides solid conceptual grounding with useful, interpretable reasoning (Reviewer eLqM).

---
During the rebuttal phase, guided by reviewers' insightful suggesitons, we have strengthened the manuscript and experiments:

**To Reviewer FHEH**: we deeply appreciate the reviewer's guidance on theoretical depth and clarity. We rewrote **Sec. 3.1.1** to ground our modules in **canonical literature** and explicitly justified their hierarchical necessity. We also incorporated more foundational work on cognitive science & world models to strengthen our theoretical framing (**Sec. 2.3**). Additionally, we added **Appendix A.10** with concrete case studies (e.g., sarcasm, indirect commands) to provide intuition. Furthermore, we improved **table captions** for self-containment, expanded the **discussion (Sec. 4.5 & 4.6)**, and addressed the other **formatting** and **citation** issues. Thank you for your meticulous attention to detail.


**To Reviewer xGU2**: we sincerely appreciate the reviewer's constructive feedback on evaluation rigor and future direction. We first addressed the concern regarding evaluation rigor by clarifying how the **Tuned baseline** effectively isolates the graph's structural benefit over mere data exposure.
We also fully agree with and appreciate the insight that **adaptive causal discovery** is a key future direction for generalizability. Crucially, we validated label generation quality by curating a **Human-verified** gold set and conducting a Round-Robin masking experiment, confirming robust label reliability against ground truth.


**To Reviewer eLqM**: we thank the reviewer for the comprehensive feedback on both writing clarity and experimental design. We addressed the concerns regarding writing clarity and experimental rigor. We improved the manuscript structure by expanding **label imputation details (Sec. 4.2)**, moving learning speed statistics to the main text **(Sec. 4.4)**, and consolidating the theory section into the **Appendix A.3** for better coherence. We also explicitly defined **causal variables (Sec. 3.1.1)**, added running example for intuition **(Fig.2 Caption)**, and clarified that our causal claims target **both** data generation and learned representation.
Experimentally, we significantly strengthened the evaluation by expanding the **LLM-judge panel** (adding Claude 3.5 Sonnet/Gemini 1.5 Pro), validating label imputation reliability via a **human-verified Round-Robin** experiment, and verifying the causal scope through a **counterfactual intervention study**.

---
We have diligently addressed all issues raised in the reviews. We firmly believe that these revisions, driven by your constructive feedback, have substantially elevated the quality and rigor of our work. We would be happy to address any remaining questions or concerns during the discussion period.

---

### Author Response · Authors · 2025-12-02
**General Response to the Area Chair**

We sincerely thank the reviewers for their time and constructive feedback. Fortunately, our rebuttal process remained **unaffected** by the recent incident, and we have comprehensively **addressed** all raised concerns in our individual responses. While the interruption prevented reviewers from replying or updating their scores, we remain grateful for their initial efforts. We also understand the significant workload and challenges this transition places on the new Area Chair. To **facilitate your efficient assessment**, we provide a concise summary of our core contributions and how we have strengthened the paper during the rebuttal phase below.

In this work, we propose ***Speech World Model (SWM)*** to advance speech **understanding** and **reasoning**. SWM constructs a **causal graph** that mimics the cognitive chain of speech perception, serving as **explicit guidance** for (multimodal) LLM reasoning. Crucially, this structure addresses **data scarcity** by enabling semi-supervised learning on partially labeled data, where gradients **propagate backward** through the edge to reinforce specific modules. Moreover, the causal graph acts as a guide for LLM finetuning, effectively pruning the language model's **search space**. This mechanism together with a casual graph not only achieves high **training efficiency** but also help mitigates **hallucinations**. We conceptualize this as a *World Model* because the graph functions essentially as a **forward dynamic model** of state-action transitions. While existing world models operate in latent vision or symbolic language spaces, we utilize a state space grounded in explicit causal graphs.

We are encouraged that the reviewers recognized the value of our **theoretical framing** and the **novelty** of uniquely integrating cognitive dimensions into a causally grounded framework (Reviewers xGU2, eLqM). They highlighted that our approach tackles a problem with considerable **real-world relevance** (Reviewer FHEH), improves learning **efficiency** via effective gradient flow (Reviewer xGU2), and provides **solid conceptual grounding** with useful, interpretable reasoning (Reviewer eLqM). Additionally, they noted that our **experiments** on various datasets provide strong evidence for the approach's **generality** (Reviewer FHEH).
Meanwhile, reviewers provided constructive suggestions regarding literature, writing clarity/layout, and supplementary experiments. In response, during the rebuttal period, we conducted a series of **supplementary experiments** to strengthen our empirical results. Furthermore, we refined the manuscript's **writing** and layout and integrated **extensive literature** to better justify and ground our core concepts.

----
**Theoretical grounding & Literature**

* **Strengthened theoretical foundations**: We significantly revised **Sec. 3.1.1** to rigorously justify our module selection (WMA, ToM, SA, Prag). We incorporated canonical citations from cognitive science and linguistics to ensure each component is historically and theoretically grounded.
* **Comprehensive review on ToM & World Models**: Addressing the need for clearer system definitions, we expanded **Sec. 2.3** and the ToM part in **Sec. 3.1.1**. These updates offer solid grounding for Theory of Mind in the context of LLMs and the interaction of World Model and cognitive science.

**Writing Clarity, layout & presentation**
* **Strategic restrcuturing**: To overcome space constraints and present a holistic theoretical framework, we consolidated the detailed derivations and theoretical arguments into **Appendix A3**. This allows for a more coherent and connected presentation of our core theory.
* **Enriched analysis & details**: Leveraging the space gained in the main text, we deepened the experimental **analysis** in **Sec. 4.5** and **4.6**. We also expanded **Sec. 4.2** to clarify the **label imputation** process and integrated **training speed** results into **Sec. 4.4** to highlight the model's efficiency.
* **Enhanced readability**: We revised the captions for **Tables 1 and 3** to make them self-contained. Additionally, we rewrote the caption for **Fig. 2** to include a **running example**, aiding reader intuition.
* **Concrete case studies**: To provide tangible evidence of our method's effectiveness, we added **Appendix A.10**, which presents concrete case studies comparing SWM against the strong baseline Voxtral.

---

> ### Author Response · Authors · 2025-12-02
> **General Response to the Area Chair (cont'd)**
>
> **Supplementary Experiments**
> * **Validated label imputation reliability**: we curated a **human-verified gold set** and employed a Round-Robin masking strategy: iteratively masking targets to test recovery via causal context. The strong performance (**accuracy/Macro-F1**) confirms that our imputation pipeline effectively leverages cross-module dependencies for label-imputation.
> * **Robust multi-judge evaluation**: We expanded our evaluation framework to include a diverse panel of judges (GPT-4o (original), +Claude 3.5 Sonnet, +Gemini 1.5 Pro). Results show exceptional **consistency** (Pearson correlation > 0.95) across judges. Crucially, SWM achieves a 100% **win-rate** against the tuned baseline across the entire panel, and SWM-Qwen2-Audio secures a 100% reasoning win-rate even against GPT-4o.
> * **Verified causal-faithfulness**: We designed a systematic **causal intervention study** (positive and negative controls) to test if the model truly reasons. The multimodal SWM demonstrates high sensitivity to causal changes while maintaining superior robustness against irrelevant perturbations. This confirms that the audio modality acts as a *reality anchor* preventing the model from blindly following false graph prompts. We also provide **two examples** from SWM (Llama3.1-8b) of the interventions in the response.
>
> -----
> Further details can be found in our individual replies to each reviewer. We have made every effort to rigorously address all feedback during this process. Ultimately, we firmly believe that **cognitive-inspired world models** and **causal LLM reasoning** represent a pivotal direction for the field. Our work is built upon this vision, and we genuinely hope to contribute these insights to foster future advancements in the community.
>
>
> Sincerely,
>
> The Authors

---

### Meta-Review · Area_Chair_aByQ · 2026-01-04

**Summary:**

This paper introduces the Speech World Model (SWM), which frames speech understanding as causal state–action planning over modular cognitive components connected by an explicit causal graph. Reviewers appreciated the strong theoretical motivation, interpretability, and empirical gains under sparse supervision. Key concerns centered on the reliance on manually specified causal structures, limited isolation of the causal graph’s contribution, and the use of LLM-generated pseudo-labels. The rebuttal substantially addressed these issues through strengthened theoretical grounding and additional validation experiments.

**Reviewer Concerns:**

Reviewers raised concerns regarding the reliance on manually specified causal graph structures, the reliability of LLM-generated pseudo-labels, and the sufficiency of experimental validation for the proposed causal formulation. In the rebuttal, the authors addressed these issues by strengthening the theoretical grounding and related work, clarifying the cognitive and communicative motivations of each module, and adding supplementary experiments, including human-verified gold data, Round-Robin masking, causal intervention studies, and expanded qualitative analyses. These additions substantially mitigate the original concerns, although questions about cross-domain generalization and adaptive causal structures remain open.

**Reviewer Scores:**

In light of the additional experiments and clarifications provided in the rebuttal, reviewers’ confidence in the method’s effectiveness and theoretical motivation has been further strengthened. Reviewers FHEH and eLqM, who initially assigned scores of 4, may consider increasing their scores, while reviewer xGU2, who gave a score of 8, is likely to maintain that score.

---

### Decision · Program_Chairs · 2026-01-26

Accept (Poster)